# Analysis of the Discrete Theory of Radiative Transfer in the Coupled "Ocean–Atmosphere" System: Current Status, Problems and Development Prospects

**Viktor P. Afanas'ev** [1], **Alexander Yu. Basov** [1], **Vladimir P. Budak** [1,*], **Dmitry S. Efremenko** [2] and **Alexander A. Kokhanovsky** [3]

1   Light Engineering Dept., National Research University "MPEI", 111250 Moscow, Russia;
    v.af@mail.ru (V.P.A.); callia12@rambler.ru (A.Y.B.)
2   Remote Sensing Technology Institute, German Aerospace Center (DLR), 82234 Oberpfaffenhofen, Germany;
    efremenkods@gmail.com
3   Vitrociset Belgium SPRL, 64289 Darmstadt, Germany; a.kokhanovsky@vitrocisetbelgium.com
*   Correspondence: budakvp@gmail.com

**Abstract:** In this paper, we analyze the current state of the discrete theory of radiative transfer. One-dimensional, three-dimensional and stochastic radiative transfer models are considered. It is shown that the discrete theory provides a unique solution to the one-dimensional radiative transfer equation. All approximate solution techniques based on the discrete ordinate formalism can be derived based on the synthetic iterations, the small-angle approximation, and the matrix operator method. The possible directions for the perspective development of radiative transfer are outlined.

**Keywords:** radiative transfer; invariant imbedding; discrete ordinate method; synthetic iterations

## 1. Introduction

Radiative transfer theory is the principal method for modeling radiation propagation in the atmosphere and the ocean in the photometric ray approximation [1,2]. In this approximation, the radiation field is decomposed into a coherent part, which determines the optical characteristics of the medium, and an incoherent one, which is related to the processes of multiple light scattering and satisfies the radiative transfer equation (RTE).

The propagation of radiation in the medium is associated with numerous physical phenomena described by various physical theories. In this article, we will consider a physical model of a continuous medium in which light is absorbed and in which discrete scatterers are located. The spatial distribution of the medium and scatterers in space is generally arbitrary.

This paper is devoted to numerical methods for solving the RTE in the discrete ordinate space. Statistical modeling techniques, such as the Monte Carlo method, follow their own logic which drops out from the presented scheme.

As will be shown in this paper, all numerical methods are based on one or another way of replacing the scattering integral with a finite sum, which replaces the desired continuous brightness distribution with discrete values or a set of coefficients for the expansion of this distribution over a system of functions. Thus, the transfer equation, its solutions, and all the corollaries from it acquire a discrete matrix form. In this case, the approximation is only the replacement of the integral by the sum, and all other conclusions can be made strictly analytically. In fact, we can talk about a discrete transfer theory. Kolmogorov A.N. [3] pointed out that with the development of modern computer technology in many cases, it is reasonable to study real phenomena, avoiding the intermediate stage of their stylization in the spirit of representations of mathematics of the infinite and continuous, moving directly to discrete models.

Today the radiative transfer theory is not only an integral branch of atmospheric physics but also a cornerstone of Earth observation technology. An analysis of modern literature over the past ten years on radiative transfer reveals a significant predominance of publications, in which different radiative transfer codes have been intercompared. In [4], the results of different radiative transfer codes agree up to the 5th digit. Such an agreement cannot be accidental and suggests that all codes presumably exploit the same method for solving the RTE.

In this regard, the state of the radiative transfer theory is very similar to the state of physics by the end of the 19th century, which was analyzed by W. Thomson (Lord Kelvin) [5]. He not only reviewed the state of physics at the turn of the century but most importantly, he highlighted two clouds (problems) that "obscure the beauty and clearness of the whole theory". These clouds were (a) the independence of the speed of light on the reference frame in the wave equation resulting from Maxwell's equations and (b) the inability to describe the dependence of blackbody radiation in the framework of classical thermodynamics. Later, these "clouds" led to the creation of quantum physics, which radically changed the whole physical paradigm of the world. Our goal in this paper is to analyze the current state of the discrete theory of radiative transfer and find its main problems, i.e., "Thomson clouds", which will determine its future development.

## 2. Boundary Value Problems of the Radiation Transfer Theory

Since the Earth radius is two orders of magnitude larger than the scales of the troposphere (which contains 75% of the atmosphere's mass) and the ocean, most radiative transfer models used in Earth remote sensing are one-dimensional. The atmosphere is treated as a set of homogeneous slabs. The horizontal inhomogeneity is considered by using the so-called independent pixel approximation (IPA) [6]. However, one-dimensional models cannot be applied at large viewing or solar zenith angles, as well as for limb observations. Moreover, in the ocean–atmosphere systems, the point sources are practically important, e.g., beacons, searchlights and lasers, which are widely used in navigation and communication systems. Cloud and surface inhomogeneities can lead to a bias in the radiances computed within one-dimensional radiative transfer models [7,8]. Such cases are the subjects for three-dimensional radiative transfer models.

Another important type of problem, which is not covered by traditional one-dimensional models is radiation transfer in stochastic media. Usually, because of mathematical difficulties, the excitements or stochastic processes in the medium are uncorrelated. Such simplifications do not consider many critical phenomena, such as stochastic lenses, which under certain circumstances, allow astronauts to see grains of sand on the ocean floor [9].

## 3. Architecture of the Boundary Value Problems in Radiative Transfer

We consider the slab of turbid medium illuminated by the plane-parallel solar beam. The direction of incidence is given by $\hat{\mathbf{l}}_0 = \left\{ \sqrt{1-\mu_0^2}, 0, \mu_0 \right\}$, $\mu_0 = (\hat{\mathbf{z}}, \hat{\mathbf{l}}_0) = \cos \vartheta_0$, where $\mu_0$ is the solar zenith angle cosine. The boundary value problem (BVP) reads as follows:

$$\begin{cases} (\hat{\mathbf{l}}, \nabla)L(z, \mathbf{r}, \hat{\mathbf{l}}) + \varepsilon(z)L(z, \mathbf{r}, \hat{\mathbf{l}}) = \frac{\sigma(z)}{4\pi} \oint x(z; \hat{\mathbf{l}}, \hat{\mathbf{l}}')L(z, \mathbf{r}, \hat{\mathbf{l}}')d\hat{\mathbf{l}}', \\ L(z, \mathbf{r}, \hat{\mathbf{l}})\Big|_{z=0, (\hat{\mathbf{z}}, \hat{\mathbf{l}}) \geq 0} = \delta(\hat{\mathbf{l}} - \hat{\mathbf{l}}_0), \quad L(z, \mathbf{r}, \hat{\mathbf{l}})\Big|_{z=d, (\hat{\mathbf{z}}, \hat{\mathbf{l}}) \leq 0} = \frac{\rho(\mathbf{r})}{\pi} \int\limits_{(\hat{\mathbf{z}}, \hat{\mathbf{l}}') \geq 0} L(z, \mathbf{r}, \hat{\mathbf{l}}')(\hat{\mathbf{z}}, \hat{\mathbf{l}}')d\hat{\mathbf{l}}'. \end{cases} \quad (1)$$

where $L(\tau, \mu, \varphi)$ is the radiance field in the viewing direction $\hat{\mathbf{l}} = \left\{ \sqrt{1-\mu^2} \cos \varphi, \sqrt{1-\mu^2} \sin \varphi, \mu \right\}$, $\mu = (\hat{\mathbf{z}}, \hat{\mathbf{l}})$, $\tau = \int\limits_0^z \varepsilon(\zeta)d\zeta$ is the optical depth, $\tau_0$ is the optical thickness of the slab, $x(\hat{\mathbf{l}}, \hat{\mathbf{l}}')$ is the single scattering phase function, $\Lambda$ is the single scattering albedo, while $\rho(\mathbf{r})$ is the reflection coefficient of

the surface. The inelastic scattering processes changing the wavelength of the light are not included. The BVP (1) is defined in the Cartesian coordinate system OXYZ, in which the OZ axis is directed downward perpendicular to the layer boundary, $\hat{\mathbf{z}}$ is a unit vector along OZ. The upper boundary of the layer is located at $z = 0$. Throughout the paper, the unit vectors are denoted with a "^" sign, while column vectors, row vectors and matrices are marked with the right arrows, left arrows and double arrows respectively.

Essentially, the BVP (1) is a set of integral equations, which consists of the RTE itself, and boundary condition, in which the reflected radiance is defined via the transmitted one. These equations are linear with respect to $L(\tau,\mu,\varphi)$. Hence, the solution can be expressed as the sum of the basic solutions according to the principle of superposition. Such a technique was considered in [10] and is referred to as the method of decomposition of BVPs. This method allows us to substitute the initial problem with boundary conditions with the hierarchy of BVPs that do not contain equations in boundary conditions. Also, the fundamental problem is formulated, through which the solutions to all problems from the hierarchy can be found. Our analysis then is focused on the fundamental problem.

Following [10], the solution to Equation (1) can be expressed as follows:

$$L(z,\mathbf{r},\hat{\mathbf{l}}) = D(z,\mathbf{r},\hat{\mathbf{l}}) + L_S(z,\mathbf{r},\hat{\mathbf{l}}), \tag{2}$$

where the "haze" component $D(z,\mathbf{r},\hat{\mathbf{l}})$ (sometimes referred to as the diffuse component) and the signal from the Lambertian surface $L_S(z,\mathbf{r},\hat{\mathbf{l}})$ satisfy the radiative transfer equation from (1) and the following boundary conditions:

$$\begin{cases} D(0,\mathbf{r},\hat{\mathbf{l}}) = \delta(\hat{\mathbf{l}} - \hat{\mathbf{l}}_0); \\ D(z_0,\mathbf{r},\hat{\mathbf{l}}) = 0; \end{cases} \text{, and} \tag{3}$$

$$\begin{cases} L_S(0,\mathbf{r},\hat{\mathbf{l}}) = 0; \\ L_S(z_0,\mathbf{r},\hat{\mathbf{l}}) = \rho R(L_S + D). \end{cases} \tag{4}$$

Here $RL(z,\mathbf{r},\hat{\mathbf{l}}') \equiv \frac{1}{\pi} \int\limits_{(\hat{\mathbf{z}},\hat{\mathbf{l}}') \geq 0} L(z,\mathbf{r},\hat{\mathbf{l}}')(\hat{\mathbf{z}},\hat{\mathbf{l}}')d\hat{\mathbf{l}}'$ is the Lambertian reflection operator. The "haze" component $D(z,\mathbf{r},\hat{\mathbf{l}})$ is related only to the scattering events in the atmosphere, while the Lambertian component $L_S(z,\mathbf{r},\hat{\mathbf{l}})$ comprises the radiance, which is reflected by the surface.

The surface reflection can be written as:

$$\rho(\mathbf{r}) = \overline{\rho} + \widetilde{\rho}(\mathbf{r}), \tag{5}$$

where $\overline{\rho} = \frac{1}{S} \int\limits_{(S)} \rho(\mathbf{r})d^2r$ is the reflection coefficient from the surface, while $S$ is the surface area.

The BVP (4) can be analyzed in the framework of the perturbation theory, namely:

$$\int |\widetilde{\rho}(\mathbf{r})|d^2r << \rho S: \quad \rho = \overline{\rho} + \xi\widetilde{\rho}, \quad L_S = \sum_{n=0}^{\infty} \xi^n L^{(n)}, \tag{6}$$

where $\xi$ is formally a small parameter, which in the final expressions is to be set $\xi = 1$. Substituting Equation (6) into Equation (4) gives:

$$\sum_{n=0}^{\infty} \xi^n L_2^{(n)} = (\overline{\rho} + \xi\widetilde{\rho})R\left(\sum_{n=0}^{\infty} \xi^n L^{(n)} + D\right), \tag{7}$$

where index 2 shows that the radiance values $L_2^{(n)}$ are taken at the surface level.

By equating the terms of Equation (7) at the same powers of $\xi$, we obtain:

$$
\begin{aligned}
&n = 0 : L_2^{(0)} = \overline{\rho}RL^{(0)} + \overline{\rho}RD; \\
&n = 1 : L_2^{(1)} = \overline{\rho}RL^{(1)} + \widetilde{\rho}R(L^{(0)} + D); \\
&n \geq 2 : L_2^{(n)} = \overline{\rho}RL^{(n)} + \widetilde{\rho}RL^{(n-1)}.
\end{aligned}
\tag{8}
$$

The signal component $L^{(0)}(z, \mathbf{r}, \overset{\wedge}{\mathbf{1}}) \equiv \overline{L}(z, \overset{\wedge}{\mathbf{1}})$ depends on the coordinate $\mathbf{r}$ and defines the radiance of the surface, which is due to the multiple reflection events from the surface $\overline{\rho}$ as well as multiple scattering events in the medium.

All Equations (8) have the same structure, namely:

$$
L_2 = \overline{\rho}RL + G(\mathbf{r}),
\tag{9}
$$

which is a convolution type equation. It is solved using the Fourier transform:

$$
\overline{\psi}(z, \mathbf{v}, \overset{\wedge}{\mathbf{1}}) = \frac{1}{2\pi} \int L(z, \mathbf{r}, \overset{\wedge}{\mathbf{1}}) e^{-i\mathbf{v}\mathbf{r}} d^2 r \equiv F\{L\}, \quad G(\mathbf{v}) = F\{G\}.
\tag{10}
$$

After the Fourier transform in Equation (10), Equation (9) takes the form:

$$
\overline{\psi}_2 = \overline{\rho}R\overline{\psi} + G(\mathbf{v}).
\tag{11}
$$

We define a new function $\psi(z, \mathbf{v}, \overset{\wedge}{\mathbf{1}})$ as such that the relation holds:

$$
\overline{\psi}(z, \mathbf{v}, \overset{\wedge}{\mathbf{1}}) = \left(\overline{\rho}R\overline{\psi} + G(\mathbf{v})\right)\psi(z, \mathbf{v}, \overset{\wedge}{\mathbf{1}}).
\tag{12}
$$

The corresponding BVP for it has the form:

$$
\begin{cases}
\psi_1 = 0, \\
\psi_2 = 1;
\end{cases}
\tag{13}
$$

which corresponds to the RTE solution for a point diffuse (PD) source:

$$
e(z; \mathbf{r}' \to \mathbf{r}) = F^{-1}\{\psi\}.
\tag{14}
$$

Substituting the obtained result in Equations (10) and (12), after transformations and the inverse Fourier transform, one reaches a solution of the BVP (Equation (9)):

$$
L(z, \mathbf{v}, \overset{\wedge}{\mathbf{1}}) = \frac{1}{2\pi} \int \frac{G(\mathbf{v})}{1 - \overline{\rho}C(\mathbf{v})} \psi(z, \mathbf{v}, \overset{\wedge}{\mathbf{1}}) e^{i\mathbf{v}\mathbf{r}} d^2 v,
\tag{15}
$$

where $C(\mathbf{v}) \equiv R\psi$ is the mean hemispherical albedo of the turbid medium slab.

Accordingly, following [10] the solution of the BVP (Equation (1)) for an arbitrary layer with a reflecting Lambertian bottom can be reduced to a superposition of solutions:

$$
L(z, \mathbf{r}, \overset{\wedge}{\mathbf{1}}) = D(z, \overset{\wedge}{\mathbf{1}}) + \overline{L}(z, \overset{\wedge}{\mathbf{1}}) + \widetilde{L}(z, \mathbf{r}, \overset{\wedge}{\mathbf{1}}) + L'(z, \mathbf{r}, \overset{\wedge}{\mathbf{1}}),
\tag{16}
$$

where $\widetilde{L}(z, \mathbf{r}, \overset{\wedge}{\mathbf{1}})$ is the radiance related to the single reflection event by the object $\widetilde{\rho}(\mathbf{r})$ and multiple scattering events from the surface, while $L'(z, \mathbf{r}, \overset{\wedge}{\mathbf{1}})$ corresponds to the nonlinear part of the radiance, which is due to multiple reflections by the object and the surface.

Thus, the solution of the BVP (Equation (1)) is reduced to the solution of the BVP for a slab with the plane parallel source, the plane diffuse source, and the point diffuse source. In [11], it was shown that any BVP of radiative transfer can be derived from the problem with the point unidirectional source.

An analysis of the solution of BVPs with concentrated sources was carried out [12]. It was shown that they always come down to solving the problem for a point unidirectional (PM) source or, in special cases, to a point isotropic source (PI). In [11,12], an analysis was performed regarding the properties of solving fundamental sources and it was shown that the angular distribution of radiance of only the source PD has no singularities, all the others have: the PM delta feature in the 0th-fold scattering term, TD, PI, and PM additionally have the inverse sine of the viewing angle in the first multiplicity, the logarithmic in the second term, and PM is the inverse sine in the third term.

## 4. The Boundary Value Problem for the Radiative Transfer Equation for a Slab

Let us analyze the numerical solution of the BVP in the case of the plane parallel source:

$$
\begin{cases}
\mu\dfrac{\partial L(\tau,\hat{1})}{\partial\tau} + L(\tau,\hat{1}) = \dfrac{\Lambda}{4\pi}\oint L(\tau,\hat{1}')x(\hat{1},\hat{1}')d\hat{1}', \\
L(\tau,\hat{1})\Big|_{\tau=0,\mu>0} = \delta(\hat{1}-\hat{1}_0), \quad L(\tau,\hat{1})\Big|_{\tau=0,\mu>0} = 0.
\end{cases}
\tag{17}
$$

Here we assume that the slab is vertically homogeneous.

To solve Equation (17) numerically, it must be discretized in the angular domain by replacing the scattering integral with a finite sum [13]. Such substitution is impossible if there are singularities in the angular distribution of the radiance field, as the singularity cannot be represented as a finite series in any basis. In [14], it was proposed to express the total radiation as a sum of the direct radiation of the source and the diffuse part. Such representation became an indispensable starting point in most solution techniques. However, all-natural objects (aerosols, clouds, etc.) contain suspended particles, which are significantly larger than the wavelength of the incident light. In accordance with the theory of G. Mie [15], the single scattering phase functions for such particles are highly forward-peaked. Consequently, in the case of highly anisotropic phase functions, replacing the integral with a finite sum can lead to significant errors [16].

## 5. Discretization of the Radiative Transfer Equation

A more advanced approach consists of the following representation of the radiance field [13]:

$$
L(\tau,\hat{1}) = L_a(\tau,\hat{1}) + \widetilde{L}(\tau,\hat{1}),
\tag{18}
$$

where $L_a(\tau,\hat{1})$ is the anisotropic part that incorporates all singularities in the angular domain, while $\widetilde{L}(\tau,\hat{1})$ is the regular part of the solution. For $\widetilde{L}(\tau,\hat{1})$ the BVP (17) is transformed into the following system [13]:

$$
\begin{cases}
\mu\dfrac{\partial\widetilde{L}(\tau,\hat{1})}{\partial\tau} + \widetilde{L}(\tau,\hat{1}) = \dfrac{\Lambda}{4\pi}\oint x(\hat{1},\hat{1}')\widetilde{L}(\tau,\hat{1}')\,d\hat{1}' + \Delta(\tau,\hat{1}); \\
\widetilde{L}(\tau,\hat{1})\Big|_{\tau=0,\ \mu>0} = 0;\ \widetilde{L}(\tau,\hat{1})\Big|_{\tau=\tau_0,\ \mu<0} = -L_a(\tau,\hat{1}),
\end{cases}
\tag{19}
$$

where the source function $\Delta(\tau,\hat{1})$ is due to the anisotropic part of the solution, namely:

$$
\Delta(\tau,\hat{1}) = -\mu\frac{dL_a(\tau,\hat{1})}{d\tau} - L_a(\tau,\hat{1}) + \frac{\Lambda}{4\pi}\oint x(\hat{1},\hat{1}')L_a(\tau,\hat{1}')d\hat{1}'.
\tag{20}
$$

Since $\widetilde{L}(\tau, \hat{\mathbf{1}})$ is a smooth function, it can be expressed on a finite basis. For instance, applying the discrete ordinate method (DOM) [17] gives:

$$\vec{\widetilde{L}}(\tau, \mu_i^\pm, \varphi) = \sum_{m=0}^{M} (2 - \delta_{0,m}) \cos(m\varphi) \vec{C}^m(\tau, \mu_i^\pm), \tag{21}$$

where $\vec{C}_\pm(\tau) \equiv \vec{C}^m(\tau, \mu_i^\pm)$, $\vec{C}(\tau) \equiv \left[ \vec{C}_-(\tau), \vec{C}_+(\tau) \right]^T$ is the column-vector of radiance values in the Fourier space along the discrete ordinate directions $\mu_j^\pm = 0.5(\zeta_j \pm 1)$, while $\zeta_j$ are the Gaussian quadrature points of the $N/2$ order. For the sake of simplicity, index $m$ is further omitted. Such discretization allows substituting the integral with a finite sum. The BVP is transformed into the matrix nonlinear differential equation of the first order with constant coefficients:

$$\frac{d\vec{C}(\tau)}{d\tau} = -\overset{\leftrightarrow}{B}\vec{C}(\tau) + \overset{\leftrightarrow}{M}{}^{-1}\vec{\Delta}(\tau), \ \overset{\leftrightarrow}{B} \equiv \overset{\leftrightarrow}{M}{}^{-1}(\overset{\leftrightarrow}{1} - \overset{\leftrightarrow}{A}\overset{\leftrightarrow}{W}), \tag{22}$$

where $w_j$ are the Gaussian quadrature weights, $P_l^n(\mu)$ are the associated Legendre polynomials, and $P_l^0(\mu) \equiv P_l(\mu)$ are the Legendre polynomials, $\overset{\leftrightarrow}{A} \equiv \sum\limits_{k=0}^{K} (2k+1)P_k^m(\mu_i^\pm)x_k P_k^m(\mu_j^\pm)$, $\overset{\leftrightarrow}{W} \equiv \frac{\Delta}{4}\begin{bmatrix} w_i & 0 \\ 0 & w_i \end{bmatrix}$, $\overset{\leftrightarrow}{M} \equiv \begin{bmatrix} \mu_i^- & 0 \\ 0 & \mu_i^+ \end{bmatrix}$, $x(\gamma) = \sum\limits_{k=0}^{\infty} \frac{2k+1}{4\pi}x_k P_k(\cos\gamma)$.

## 6. Propagators and Scatters

The general solution of the inhomogeneous Equation (22) is given as the sum of the particular solution of the inhomogeneous equation and general solution of the homogeneous equation:

$$-\vec{C}(0) + \overset{\leftrightarrow}{P}(0,\tau_0)\vec{C}(\tau_0) = \int_0^{\tau_0} \overset{\leftrightarrow}{P}(0,\tau)\overset{\leftrightarrow}{M}{}^{-1}\vec{\Delta}(\tau, \mu_0)d\tau, \tag{23}$$

where

$$\overset{\leftrightarrow}{P}(t, \tau) \equiv e^{\overset{\leftrightarrow}{B}(\tau - t)} \tag{24}$$

is the solution to the homogeneous equation. It is referred to as the propagator and relates the radiance fields at spatial coordinates $t$ and $\tau$.

Note, the propagator comprises both negative and positive exponentials, which physically corresponds to the downward and upward radiances, respectively. The positive exponentials increase the condition number of the matrix $\overset{\leftrightarrow}{P}$ making the numerical procedure unstable as the optical thickness increases. To get rid of this effect, the scaling transformation is applied [18]:

$$-\overset{\leftrightarrow}{S}\overset{\leftrightarrow}{U}{}^{-1}\vec{C}(0) + \overset{\leftrightarrow}{H}\overset{\leftrightarrow}{U}{}^{-1}\vec{C}(\tau_0) = \vec{J}, \ \vec{J} \equiv \overset{\leftrightarrow}{S}\int_0^{\tau_0} e^{\overset{\leftrightarrow}{\Gamma}t}\overset{\leftrightarrow}{U}{}^{-1}\overset{\leftrightarrow}{M}{}^{-1}\vec{\Delta}(t)dt, \tag{25}$$

where $\overset{\leftrightarrow}{U}$ is the eigenvector matrix of $\overset{\leftrightarrow}{B}$, $\overset{\leftrightarrow}{\Gamma} = \mathrm{diag}(\overset{\leftrightarrow}{\Gamma}_-, \overset{\leftrightarrow}{\Gamma}_+)$ is the diagonal matrix of positive $\overset{\leftrightarrow}{\Gamma}_+$ and negative $\overset{\leftrightarrow}{\Gamma}_-$ eigenvalues, and

$$\overset{\leftrightarrow}{B} = \overset{\leftrightarrow}{U}\overset{\leftrightarrow}{G}\overset{\leftrightarrow}{U}{}^{-1}, \tag{26}$$

$$\overset{\leftrightarrow}{\Gamma}_- = -\overset{\leftrightarrow}{\Gamma}_+; \overset{\leftrightarrow}{U}{}^{-1} \equiv \begin{bmatrix} \overset{\leftrightarrow}{u}_{11} & \overset{\leftrightarrow}{u}_{12} \\ \overset{\leftrightarrow}{u}_{21} & \overset{\leftrightarrow}{u}_{22} \end{bmatrix}, \overset{\leftrightarrow}{S} = \begin{bmatrix} \overset{\leftrightarrow}{1} & 0 \\ 0 & e^{-\overset{\leftrightarrow}{\Gamma}_+\tau_0} \end{bmatrix}, \overset{\leftrightarrow}{H} = \begin{bmatrix} e^{\overset{\leftrightarrow}{\Gamma}_-\tau_0} & 0 \\ 0 & \overset{\leftrightarrow}{1} \end{bmatrix}.$$

Note, that the BVP (17) and the corresponding Equation (22) are the two-point boundary problems, i.e., the boundary conditions are defined at two boundaries. Hence, solution (23) expressed through the propagators is not complete [19].

The column-vectors $\vec{C}_+(0)$, $\vec{C}_-(\tau_0)$ in Equation (25) correspond to the radiances defined by the boundary conditions. The column-vectors $\vec{C}_-(0)$, $\vec{C}_+(\tau_0)$ correspond to the transmitted and reflected radiances. They can be found from Equation (25):

$$
\begin{bmatrix} \vec{C}_-(0) \\ \vec{C}_+(\tau_0) \end{bmatrix} = \begin{bmatrix} \vec{F}_- \\ \vec{F}_+ \end{bmatrix} + \begin{bmatrix} \overleftrightarrow{R} & \overleftrightarrow{T} \\ \overleftrightarrow{T} & \overleftrightarrow{R} \end{bmatrix} \begin{bmatrix} \vec{C}_+(0) \\ \vec{C}_-(\tau_0) \end{bmatrix},
\tag{27}
$$

where $\begin{bmatrix} \vec{F}_- \\ \vec{F}_+ \end{bmatrix} = \overleftrightarrow{h}\,\vec{J}$, $\begin{bmatrix} \overleftrightarrow{R} & \overleftrightarrow{T} \\ \overleftrightarrow{T} & \overleftrightarrow{R} \end{bmatrix} = \overleftrightarrow{h} \begin{bmatrix} \overleftrightarrow{u}_{12} & -e^{\overrightarrow{\Gamma}-\tau_0}\overleftrightarrow{u}_{11} \\ e^{-\overrightarrow{\Gamma}+\tau_0}\overleftrightarrow{u}_{22} & -\overleftrightarrow{u}_{21} \end{bmatrix}$, $\overleftrightarrow{h} \equiv \begin{bmatrix} -\overleftrightarrow{u}_{11} & e^{\overrightarrow{\Gamma}-\tau_0}\overleftrightarrow{u}_{12} \\ -e^{-\overrightarrow{\Gamma}+\tau_0}\overleftrightarrow{u}_{21} & \overleftrightarrow{u}_{22} \end{bmatrix}^{-1}$.

The solution in the form of scatters (see Equation (27)) relates the incoming and outgoing radiances and, thus, can be considered as a generalization of the radiance coefficients. The column-vectors $\vec{F}$ correspond to the intrinsic radiation of the layer, while the matrices $\overleftrightarrow{R}$ and $\overleftrightarrow{T}$ are the discrete values of reflection and transmission coefficients, respectively [13].

Equation (27) can be transformed into the propagator-like form:

$$
\vec{C}(\tau_0) = \begin{bmatrix} \overleftrightarrow{T}_+ - \overleftrightarrow{R}_-\overleftrightarrow{T}_-^{-1}\overleftrightarrow{R}_+ & \overleftrightarrow{R}_-\overleftrightarrow{T}_-^{-1} \\ -\overleftrightarrow{T}_-^{-1}\overleftrightarrow{R}_+ & \overleftrightarrow{T}_-^{-1} \end{bmatrix} + \overleftrightarrow{F}.
\tag{28}
$$

In [20], such a kind of transformation was referred to as the stellar product. The entries of scatter matrices satisfy the Riccati equation, which corresponds to the one-point BVP [20,21].

## 7. Invariance Property of the Solution

Consider the medium consisting of two adjacent layers:

$$
\begin{bmatrix} \vec{C}_-^1 \\ \vec{C}_+^1 \end{bmatrix} = \begin{bmatrix} \vec{F}_-^1 \\ \vec{F}_+^1 \end{bmatrix} + \begin{bmatrix} \overleftrightarrow{R}_{1-} & \overleftrightarrow{T}_{1-} \\ \overleftrightarrow{T}_{1+} & \overleftrightarrow{R}_{1+} \end{bmatrix} \begin{bmatrix} \vec{C}_\downarrow^1 \\ \vec{C}_\uparrow^1 \end{bmatrix}, \begin{bmatrix} \vec{C}_-^2 \\ \vec{C}_+^2 \end{bmatrix} = \begin{bmatrix} \vec{F}_-^2 \\ \vec{F}_+^2 \end{bmatrix} + \begin{bmatrix} \overleftrightarrow{R}_{2-} & \overleftrightarrow{T}_{2-} \\ \overleftrightarrow{T}_{2+} & \overleftrightarrow{R}_{2+} \end{bmatrix} \begin{bmatrix} \vec{C}_\downarrow^2 \\ \vec{C}_\uparrow^2 \end{bmatrix},
\tag{29}
$$

where the bottom index is the layer index (i.e., 1 corresponds to the upper layer, while 2 corresponds to the bottom layer). Note that since the layers are adjacent and the solution is continuous, we have:

$$
\vec{C}_+^1 = \vec{C}_\downarrow^2 \equiv \vec{C}_\downarrow, \ \vec{C}_-^2 = \vec{C}_\uparrow^1 \equiv \vec{C}_\uparrow.
\tag{30}
$$

By expressing transmitted $\vec{C}_-^1$ and reflected $\vec{C}_+^2$ radiances by incident radiances $\vec{C}_\downarrow^1$ and $\vec{C}_\uparrow^2$, we obtain:

$$
\begin{bmatrix} \vec{C}_-^1 \\ \vec{C}_+^2 \end{bmatrix} = \begin{bmatrix} \vec{F}_-^1 + \overleftrightarrow{T}_{1-}\overleftrightarrow{\alpha}_1\left(\overleftrightarrow{R}_{2-}\vec{F}_+^1 + \vec{F}_-^2\right) \\ \vec{F}_+^2 + \overleftrightarrow{T}_{2+}\overleftrightarrow{\alpha}_2\left(\vec{F}_+^1 + \overleftrightarrow{R}_{1+}\vec{F}_-^2\right) \end{bmatrix} + \begin{bmatrix} \overleftrightarrow{R}_{1-} + \overleftrightarrow{T}_{1-}\overleftrightarrow{\alpha}_1\overleftrightarrow{R}_{2-}\overleftrightarrow{T}_{1+} & \overleftrightarrow{T}_{1-}\overleftrightarrow{\alpha}_1\overleftrightarrow{T}_{2-} \\ \overleftrightarrow{T}_{2+}\overleftrightarrow{\alpha}_2\overleftrightarrow{T}_{1+} & \overleftrightarrow{R}_{2+} + \overleftrightarrow{T}_{2+}\overleftrightarrow{\alpha}_2\overleftrightarrow{R}_{1+}\overleftrightarrow{T}_{2-} \end{bmatrix} \begin{bmatrix} \vec{C}_\downarrow^1 \\ \vec{C}_\uparrow^2 \end{bmatrix}
\tag{31}
$$

where $\overleftrightarrow{\alpha}_1 = \left(\overleftrightarrow{1} - \overleftrightarrow{R}_{2-}\overleftrightarrow{R}_{1+}\right)^{-1}$, $\overleftrightarrow{\alpha}_2 = \left(\overleftrightarrow{1} - \overleftrightarrow{R}_{1+}\overleftrightarrow{R}_{2-}\right)^{-1}$.

We can see that solution (31) for two adjacent layers is equivalent to solution (27) for a single layer with effective parameters of the medium. Namely, the invariance property of the solution is expressed in the form of scatterers, which corresponds to the discrete setting of the invariant imbedding method

of V.A. Ambartsumian [22]. Indeed, let us consider a system of two layers, where the upper layer 1 is infinitely thin with an optical thickness $d\tau$, while the lower layer 2 is semi-infinite. As layer 1 is infinitely thin, its radiance field can be derived by using the single scattering approximation. For that, we consider the Peiperl's integral equation, which is the formal solution to the radiative transfer equation, given that the scattering integral is known. For a layer illuminated by radiance $L_\uparrow(\hat{1})$ from the top, and by $L_\uparrow(\hat{1})$ from the bottom, the integral radiative transfer equation reads as:

$$
L(\tau,\hat{1}) =
\begin{cases}
L_\downarrow(\hat{1})e^{-\tau/\mu} + \frac{\Lambda}{4\pi\mu}\int_0^\tau e^{-(\tau-t)/\mu}\oint x(\hat{1},\hat{1}')L(t,\hat{1}')d\hat{1}'\,dt, & \mu \geq 0;\\[2mm]
L_\uparrow(\hat{1})e^{-(\tau_0-\tau)/\mu} - \frac{\Lambda}{4\pi\mu}\int_\tau^{\tau_0} e^{-(\tau-t)/\mu}\oint x(\hat{1},\hat{1}')L(t,\hat{1}')d\hat{1}'\,dt, & \mu < 0.
\end{cases}
\tag{32}
$$

The single scattering approximation leads to the following solution:

$$
L(0,\hat{1}) = L_\uparrow(\hat{1})e^{\tau_0/\mu} - \frac{\Lambda}{4\pi\mu}\int_0^{\tau_0}e^{t/\mu}\left[\int_{\mu>0} x(\hat{1},\hat{1}')L_\downarrow(\hat{1}')e^{-t/\mu'\hat{1}'} + \int_{\mu<0}x(\hat{1},\hat{1}')L_\uparrow(\hat{1}')e^{(\tau_0-t)/\mu'\hat{1}'}\right]dt,
$$
$$
L(\tau_0,\hat{1}) = L_\downarrow(\hat{1})e^{-\tau_0/\mu} + \frac{\Lambda}{4\pi\mu}\int_0^{\tau_0}e^{-(\tau_0-t)/\mu}\left[\int_{\mu>0} x(\hat{1},\hat{1}')L_\downarrow(\hat{1}')e^{-t/\mu'\hat{1}'} + \int_{\mu<0}x(\hat{1},\hat{1}')L_\uparrow(\hat{1}')e^{(\tau_0-t)/\mu'\hat{1}'}\right]dt,
\tag{33}
$$

where the upper equation corresponds to the downwelling radiance, and the lower equation corresponds to the upwelling radiance.

Performing the integration in Equation (33), while considering the Fourier expansion and applying the discrete ordinate method, we obtain a solution for the infinitely thin layer in the form of scatters:

$$
\begin{bmatrix}\vec{C}_-^1\\ \vec{C}_+^1\end{bmatrix} = \begin{bmatrix}-\overleftrightarrow{\mu}_+\overleftrightarrow{x}_+\overleftrightarrow{w}d\tau & (1+\overleftrightarrow{\mu}_+d\tau)-\overleftrightarrow{\mu}_+\overleftrightarrow{x}_-\overleftrightarrow{w}d\tau\\ (1-\overleftrightarrow{\mu}_+d\tau)+\overleftrightarrow{x}_-\overleftrightarrow{w}d\tau & \overleftrightarrow{\mu}_+\overleftrightarrow{x}_+\overleftrightarrow{w}d\tau\end{bmatrix}\begin{bmatrix}\vec{C}_\downarrow\\ \vec{C}_\uparrow\end{bmatrix} \equiv \begin{bmatrix}\overleftrightarrow{R}_{1-} & \overleftrightarrow{T}_{1-}\\ \overleftrightarrow{T}_{1+} & \overleftrightarrow{R}_{1+}\end{bmatrix}\begin{bmatrix}\vec{C}_\downarrow\\ \vec{C}_\uparrow\end{bmatrix}.
\tag{34}
$$

Since layer 2 is semi-infinite, we have:

$$
\overleftrightarrow{T}_{2+} = \overleftrightarrow{T}_{2-} = 0,\ \vec{C}_\uparrow^2 = 0: \vec{C}_-^1 = \left(\overleftrightarrow{R}_{1-} + \overleftrightarrow{T}_{1-}\overleftrightarrow{\alpha}_1\overleftrightarrow{R}_{2-}\overleftrightarrow{T}_{1+}\right)\vec{C}_\downarrow^1,\ \overleftrightarrow{\alpha}_1 = \left(\overleftrightarrow{1} - \overleftrightarrow{R}_{2-}\overleftrightarrow{R}_{1+}\right)^{-1}.
\tag{35}
$$

Moreover, adding the infinitely thin layer 1 to the semi-infinite layer 2 does not lead to the change in the reflection, namely:

$$
\vec{C}_-^1 = \overleftrightarrow{R}_{2-}\vec{C}_\downarrow^1
\tag{36}
$$

and Equation (31) is transformed into the following expression:

$$
\overleftrightarrow{R}_{2-} = \overleftrightarrow{R}_{1-} + \overleftrightarrow{T}_{1-}\overleftrightarrow{\alpha}_1\overleftrightarrow{R}_{2-}\overleftrightarrow{T}_{1+}.
\tag{37}
$$

The expressions for the transmission and reflection of layer 1 include an infinitely small layer thickness $d\tau$. By considering the Taylor expansion we obtain:

$$
\overleftrightarrow{\alpha}_1 = \left(\overleftrightarrow{1} - \overleftrightarrow{R}_{2-}\overleftrightarrow{R}_{1+}\right)^{-1} \approx \overleftrightarrow{1} + \overleftrightarrow{R}_{2-}\overleftrightarrow{R}_{1+} = \overleftrightarrow{1} + \overleftrightarrow{R}_\infty\overleftrightarrow{x}_+\overleftrightarrow{w}d\tau,\ \overleftrightarrow{R}_{2-} \equiv \overleftrightarrow{R}_\infty.
\tag{38}
$$

Then, substituting reflection matrices corresponding to Equation (34) into Equation (37) gives:

$$
\overleftrightarrow{\mu}_+\overleftrightarrow{R}_\infty - \overleftrightarrow{R}_\infty\overleftrightarrow{\mu}_+ = \overleftrightarrow{\mu}_+\overleftrightarrow{x}_+\overleftrightarrow{w} + \overleftrightarrow{\mu}_+\overleftrightarrow{x}_-\overleftrightarrow{w}\overleftrightarrow{R}_\infty - \overleftrightarrow{R}_\infty\overleftrightarrow{x}_-\overleftrightarrow{w} - \overleftrightarrow{R}_\infty\overleftrightarrow{x}_+\overleftrightarrow{w}\overleftrightarrow{R}_\infty
\tag{39}
$$

which corresponds to the expression derived by V.A. Ambartsumian [22] but in a discrete form.

The invariance principle can be applied to a layer of arbitrary vertical inhomogeneity, by breaking it into an arbitrary number of homogeneous layers. In this case, two adjacent layers can be replaced by one layer described by Equation (31). This approach in the radiative transfer theory is called the matrix operator method [23] (MOM). Note that in this paper Equation (31) is derived for the arbitrary anisotropic part of the radiance field (see Equation (18)).

## 8. The Anisotropic Part of the Solution

The accuracy of solution (11) as well as the computational time is governed by the number of discrete ordinates *N*, the number of azimuthal modes *M* and the number of phase function expansion coefficients, *K*. Generally, $M \approx N \approx K$ [13]. However, by appropriate choice of the anisotropic part in Equation (18), the regular part is close to an isotropic function, and the number of discrete ordinates and the number of azimuthal modes can be substantially reduced. In this case, $K >> N >> M$. Since the computational time increases *M* and *N*, the reduction of *M* and *N* leads to the performance enhancement.

A question arises: how to choose the anisotropic part in the best way? To answer to this question, we expand the radiance field in the Legendre series:

$$L_a(\tau, \mathbf{v}) = \sum_{k=1}^{\infty} \frac{2k+1}{4\pi} Z_k(\tau) P_k(\mathbf{v}). \tag{40}$$

Note that the spectrum $Z_k(\tau)$ of the anisotropic part is a smooth function of *k*, while $Z_k(\tau)$ drops fast if $L_a(\tau, \mathbf{v})$ is a forward peaked function. Hence, for the anisotropic part we can introduce a continuous function $Z(k, \tau)$, which coincides with $Z_k(\tau)$ at integer points *k*, i.e., $Z(k, \tau) = Z_k(\tau)$. Next, we consider the Taylor expansion of $Z(k, \tau)$:

$$Z(k \pm 1, \tau) \simeq Z_k(\tau) \pm \left. \frac{\partial Z(\mathrm{k}, \tau)}{\partial \mathrm{k}} \right|_{\mathrm{k}=k}. \tag{41}$$

Substituting Equation (41) into the BVP for $L_a$ and applying the spherical harmonics method one can obtain the following expression [24]:

$$Z_k(\tau) \equiv \exp(-(1 - \Lambda x_k)\tau/\mu_0), \tag{42}$$

which is referred to as the small angle modification of the spherical harmonics method. In [4] it was reported that such a technique accelerates computations of the radiance field by two orders of magnitude. The algorithm based on the allocation of the anisotropic solution on the basis of expression (42) received the name of the modified DOM (MDOM) [4].

We estimate the accuracy of the MDOM based on a comparison with the exact analytical solution of S. Chandrasekhar [25] for the radiance of reflected radiation from a semi-infinite medium with a Rayleigh scattering phase function. We take the functions necessary for calculating the exact solution from the tables [26]. The relative error of the MDOM relative to the exact analytical solution $\Delta$ for a medium with $\Lambda = 0.999$ and two incidence angles $\vartheta_0 = 0°$ and $70°$ for different sight angles is shown in Figure 1. The observation scheme is adopted, in which the direction to the Nadir $\vartheta = 180°$. As one can see, the error does not exceed $10^{-5}$ in almost the entire range of sight angles.

Here it is necessary to note the considerable role that accurate analytical methods play in the transfer theory, but this is a vast separate topic that goes beyond the scope of our article. Note that the development of the analytical approach did not stop at the classics of the transfer theory but is still actively continuing today, see, for example [27].

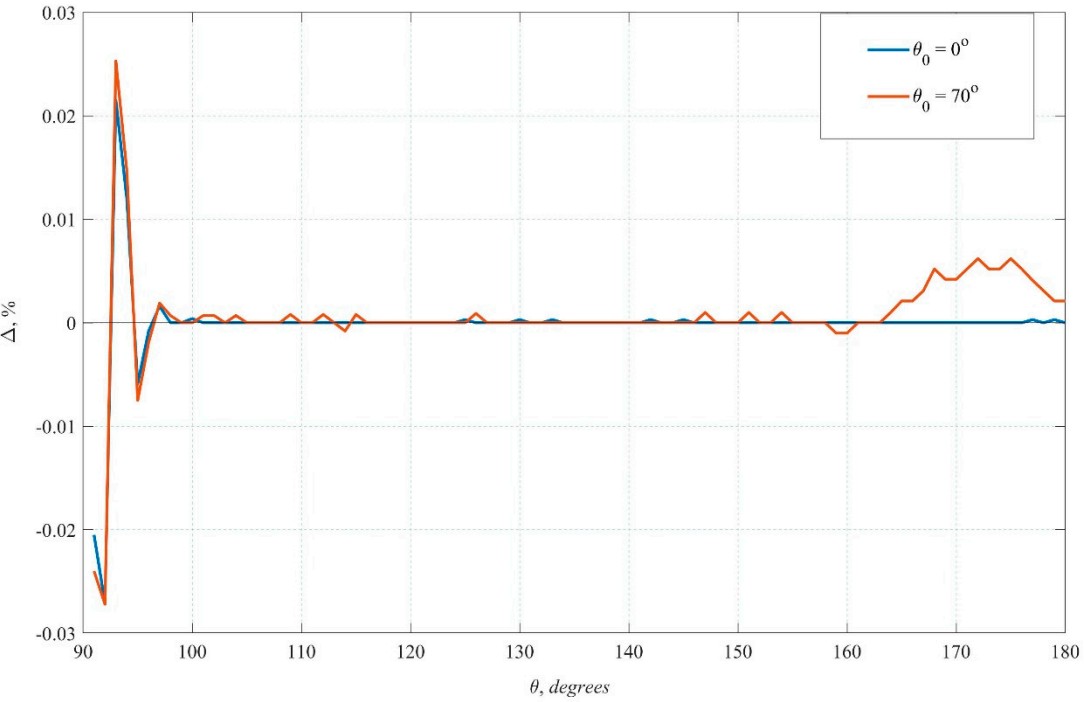

**Figure 1.** The relative error of the MDOM to the exact RTE solution for a semi-infinite layer of a cloudy medium with Rayleigh scattering.

## 9. Reflection and Refraction at the Ocean–Atmosphere Interface

Let us consider a layer illuminated by a plane unidirectional source with a diffusely reflecting substrate at the bottom with the reflection coefficient $\rho$. In this case, the boundary condition at the bottom in Equation (17) reads as follows:

$$L(\tau_0, \hat{\mathbf{1}})\Big|_{\mu<0} = \frac{\rho}{\pi} \int_{(\mu>0)} L(\tau_0, \hat{\mathbf{1}}')\mu d\hat{\mathbf{1}}'. \tag{43}$$

For Fourier expansion coefficients in the discrete ordinate space, Equation (43) can be written in the matrix form as:

$$m = 0: \ \vec{C}_-(\tau_0) = 2\rho \overleftrightarrow{R}_L \vec{C}_+(\tau_0); \ \forall m > 0: \ \vec{C}_-(\tau_0) = \vec{0}, \tag{44}$$

where $\overleftrightarrow{R}_L$ is the Lambertian reflection matrix consisting of $N/2$ identical rows $\{\mu_j^+ w_j\}$. According to Equation (31) for the zeroth azimuthal mode we have:

$$\vec{C}_-(0) = \vec{F}_- + 2\rho \overleftrightarrow{T}_- \left(\overleftrightarrow{1} - 2\rho \overleftrightarrow{R}_L \overleftrightarrow{R}\right)^{-1} \overleftrightarrow{R}_L \vec{F}_+. \tag{45}$$

However, this approach cannot be directly applied at the boundary with refraction, since the directions of the ordinates ("rays") change at the boundary according to Snell's law. The solution to this problem was proposed in [28]. We consider the coupled ocean–atmosphere system. For simplicity, we set the refractive index of the atmosphere to 1, while the reflective index of the ocean is $n_o > 1$. The discrete ordinates in the atmosphere are related to those in the ocean through Snell's law as follows:

$$\mu_a = \sqrt{1 - n_o^2(1 - \mu_o^2)}. \tag{46}$$

These oceanic ordinates belong to the refraction zone. For $\mu_o < \mu_t \equiv \sqrt{1 - 1/n_o^2}$ (where $\mu_t$ is the total reflection angle cosine) we have the total reflection zone, in which the rays cannot escape from the water body, but rather are reflected from the water-air interface. The boundary condition at the ocean–atmosphere interface reads as follows:

$$\int\limits_{-1}^{1} Q_k^m(\mu')C^m(\tau,\mu')d\mu' = \int\limits_{-1}^{-\mu_t} Q_k^m(\mu')C^m(\tau,\mu')d\mu' + \int\limits_{-\mu_t}^{\mu_t} Q_k^m(\mu')C^m(\tau,\mu')d\mu' + \int\limits_{\mu_t}^{1} Q_k^m(\mu')C^m(\tau,\mu')d\mu'. \tag{47}$$

The first and last integrals are related to the refraction zone, while the second integral corresponds to the total reflection zone. After performing the change of variables in the second integral:

$$\int\limits_{-\mu_t}^{\mu_t} Q_k^m(\mu')C^m(\tau,\mu')d\mu' = \int\limits_{-1}^{1} Q_k^m(\nu)C^m(\tau,\nu)d\nu, \tag{48}$$

where $\mu' = \mu_t \nu$, the Gaussian quadrature formula can be applied with $N_t$ points per hemisphere. Consequently, we obtain the two column-vectors $\vec{C}_+^t$, $\vec{C}_-^t$ of downwelling and upwelling radiance values, respectively, in the Fourier space along discrete ordinate directions. Note that in the case of the flat oceanic surface $\vec{C}_+^t$ and $\vec{C}_-^t$ are related to each other by the law of specular reflection. For the first and last integrals in Equation (47) we perform the change of variables according to Equation (46):

$$\mu_o = \sqrt{1 - (1 - \mu_a^2)/n_o^2}, \quad d\mu_o = \mu_a d\mu_a / \sqrt{n_o^2 - (1 - \mu_a^2)}. \tag{49}$$

Thus, a complete correspondence is established between atmospheric $\vec{C}_+^a$, $\vec{C}_-^a$ and ocean $\vec{C}_+^o$, $\vec{C}_-^o$ ordinates. For vectors $\vec{C}_+^{ocn} = \left[\vec{C}_+^t; \vec{C}_+^o\right]$, $\vec{C}_-^{ocn} = \left[\vec{C}_-^o; \vec{C}_-^t\right]$, where square brackets denote the concatenation of vectors, the matrix operator method gives the following equation:

$$\begin{bmatrix} \vec{C}_-^a \\ \vec{C}_+^{ocn} \end{bmatrix} = \begin{bmatrix} \overleftrightarrow{R} & \overleftrightarrow{T}_{ao} \\ \overleftrightarrow{T}_{oa} & \overleftrightarrow{R}_{oo} \end{bmatrix} \begin{bmatrix} \vec{C}_+^a \\ \vec{C}_-^{ocn} \end{bmatrix}, \quad \overleftrightarrow{T}_{ao} \equiv \begin{bmatrix} \overleftrightarrow{T} & 0 \end{bmatrix}, \quad \overleftrightarrow{T}_{oa} \equiv \begin{bmatrix} 0 \\ \overleftrightarrow{T} \end{bmatrix}, \quad \overleftrightarrow{R}_{oo} \equiv \begin{bmatrix} 0 & 1 \\ \overleftrightarrow{R} & 0 \end{bmatrix} \tag{50}$$

where $\overleftrightarrow{R}$, $\overleftrightarrow{T}$ are the Fresnel reflection and transmission matrices, respectively.

## 10. Synthetic Iteration Method

In [29,30] the performance assessment of MDOM was conducted. In particular, the computational time of MDOM was measured for computing the angular distributions of the reflected radiances with different values of $N$ and $M$. At the same time, it was found that for all $N$ and $M$ the computed radiances coincide in the entire range of angles except for the small glory region. In other words, insufficient numbers of discrete ordinates and azimuthal harmonics $N$ and $M$ affects only a narrow range of angles. The calculation in the glory region required a 100 times increase of $N$ and $M$, which led to the increase of the computational time by more than 300 times. Thus, the computations of the radiance field in the wide angular range is two orders of magnitude faster than computations of sharp peaks. Why is it so?

Let us consider the Legendre expansion of the radiance field:

$$L_m(\tau,\mu) = \sum_{k=1}^{N} \frac{2k+1}{2} L_k^m P_k(\mu). \tag{51}$$

Legendre polynomials of an order not higher than $N$ can be expressed through polynomials of order $N + 1$ by the discrete samples as follows:

$$P_k(\mu) = \sum_{i=1}^{N+1} P_k(\mu_i) \frac{P_{N+1}(\mu)}{(\mu - \mu_i)P'_{N+1}(\mu_i)}, \tag{52}$$

where $\mu_i$ are the roots of $P_{N+1}(\mu)$. Substituting Equation (52) into Equation (51) gives:

$$L_m(\tau, \mu) = \sum_{i=1}^{N+1} \frac{P_{N+1}(\mu)}{(\mu - \mu_i)P'_{N+1}(\mu_i)} \sum_{k=0}^{N} \frac{2k+1}{2} L_k^m P_k(\mu_i) = \sum_{i=1}^{N+1} L_m(\tau, \mu_i) \frac{P_{N+1}(\mu)}{(\mu - \mu_i)P'_{N+1}(\mu_i)}. \tag{53}$$

This is the Lagrange interpolation formula for $L_m(\tau, \mu)$. These relations can be regarded as analogous to the Nyquist-Shannon-Kotelnikov sampling theorem. The following comments are in order:

1)  Since the separation of the anisotropic part of the solution provides the most accurate discrete representation of the scattering integral, the MDOM determines the mean convergence;
2)  The convergence of the solution in the uniform metric is determined by the features of the angular distribution of the scattering phase function, then the convergence here of any method for isolating anisotropy will be equivalent;
3)  To achieve good convergence in a uniform metric, the sampling interval must correspond to the angular size of the finest detail of the radiance distribution necessary to solve a practical problem.

Thus, the Nyquist-Shannon-Kotelnikov sampling theorem put a lower constraint on the number of discrete ordinates *N*. Moreover, *N* is the main parameter that governs the accuracy and computational time. To accelerate the calculation, *N* should be kept as low as possible. The synthetic iteration method allows for the overcoming of the limit defined by the Nyquist-Shannon-Kotelnikov sampling theorem and achieves performance enhancement without compromising accuracy.

The method of synthetic iterations (SI) was initially proposed in nuclear physics [31]. It consists of two steps. In the first step, an approximate solution is sought that converges well in the average energy metric. In the second step, the iteration is performed, which refines the solution and significantly increases the convergence in a uniform angular metric. Since MDOM convergences fast in the average metric, one can expect a significant increase in convergence rate after a single iteration [32].

A numerical comparison of the reflected radiance after the first iteration of MDOM is given in [29,30]. To calculate the glory region in the MDOM program, $N = 801$ and $M = 256$ are required, which corresponds to a sampling step of less than 0.5°. To achieve the same accuracy in the synthetic iteration, only $N = 11$, $M = 4$ are required, which reduces the computation time by almost 60 times. Accordingly, the synthetic iteration from MDOM allows one to calculate the angular distribution of radiance in one second with the relative error in the uniform metric less than 1%.

## 11. Polarized Radiation Transfer

The considered scalar transfer theory allows a simple generalization to the vector case of polarization. In the ray approximation, the polarized radiation is described [13] by the vector of Stokes parameters $\vec{L} = [I, Q, U, V]^T$. The BVP of the vectorial RTE (VRTE) for $\vec{L}$ is similar to (1) and reads as follows:

$$\begin{cases} \mu \dfrac{\partial \vec{L}(\tau, \hat{1})}{\partial \tau} + \vec{L}(\tau, \hat{1}) = \dfrac{A}{4\pi} \oint \overset{\leftrightarrow}{R}(\hat{1} \times \hat{1}' \to \hat{1}_0 \times \hat{z}) \overset{\leftrightarrow}{x}(\hat{1}\hat{1}') \overset{\leftrightarrow}{R}(\hat{1}_0 \times \hat{z} \to \hat{1} \times \hat{1}') \vec{L}(\tau, \hat{1}') d\hat{1}' \, ; \\ \vec{L}(0, \hat{1})\Big|_{\mu \geq 0} = \vec{L}_0 \delta(\hat{1} - \hat{1}_0); \ \vec{L}(\tau_0, \hat{1})\Big|_{\mu \leq 0} = \vec{0}, \end{cases} \tag{54}$$

where $\overset{\leftrightarrow}{R}(\cdot)$ is the rotator with respect to the dihedral angle between the $\hat{1} \times \hat{1}'$ and $\hat{1}_0 \times \hat{z}$ planes, $\overset{\leftrightarrow}{x}(\hat{1}\hat{1}')$ is the scattering matrix, $\vec{L}_0$ is the Stokes vector for the top boundary condition.

Discretization of the VRTE by DOM is based on reducing the double integral to the single one by applying the addition theorem. In the case of polarization, the scattering matrix is surrounded

by the rotator matrices $\overset{\leftrightarrow}{R}(\cdot)$ which disturb the azimuthal symmetry and do not allow using the addition theorem for the surface harmonics. Kuščer-Ribarič [32] applied the circular basis to determine polarization. This form is called the CP-representation (circular polarization) and can be obtained from the real-number energetic SP-representation (Stokes polarization) using the following matrix transformation:

$$
\overset{\rightarrow}{L}_{CP} \equiv \begin{bmatrix} L_{+2} \\ L_{+0} \\ L_{-0} \\ L_{-2} \end{bmatrix} = \frac{1}{2} \begin{bmatrix} 0 & 1 & -i & 0 \\ 1 & 0 & 0 & -1 \\ 1 & 0 & 0 & 1 \\ 0 & 1 & i & 0 \end{bmatrix} \begin{bmatrix} I \\ Q \\ U \\ V \end{bmatrix} \equiv \overset{\leftrightarrow}{T}_{CS} \overset{\rightarrow}{L}_{SP}, \quad \overset{\leftrightarrow}{T}_{SC} = \overset{\leftrightarrow}{T}_{CS}^{-1}, \tag{55}
$$

where *i* is the imaginary number. Thus, CP-representation is the complex basis, but it makes possible to diagonalize the rotation matrix:

$$
\overset{\leftrightarrow}{R}_{CP}(\chi) = \overset{\leftrightarrow}{T}_{CS} \overset{\leftrightarrow}{R}(\chi) \overset{\leftrightarrow}{T}_{SC}^{-1} = \mathrm{diag}\begin{bmatrix} e^{+i2\chi} & e^{+i0\chi} & e^{-i0\chi} & e^{-i2\chi} \end{bmatrix}. \tag{56}
$$

It changes the rotator form and allows using generalized surface functions $P_{mn}^{k}(\cos\theta)$ for the scattering matrix representation, for which the special form of the addition theorem [33] can be applied:

$$
e^{-im\chi} P_{mn}^{k}(\hat{\mathbf{1}} \cdot \hat{\mathbf{1}}') e^{-in\chi'} = \sum_{q=-k}^{k} (-1)^{q} P_{mn}^{k}(\hat{\mathbf{1}} \cdot \hat{\mathbf{z}}) P_{mn}^{k}(\hat{\mathbf{z}} \cdot \hat{\mathbf{1}}') e^{iq(\varphi-\varphi')}, \tag{57}
$$

where $\varphi$ and $\varphi'$ are the azimuth of $\hat{\mathbf{1}}$ and $\hat{\mathbf{1}}'$ about an axis *OZ*, respectively.

As a result, all the coefficients in the VRTE become complex numbers. After applying the addition theorem for the generalized spherical function, one should return to the Stokes polarization (SP). For the column vector defined as:

$$
\overset{\rightarrow}{C}_{\pm}(\tau) = \left[ I(\tau,\mu_{1}^{\pm}), Q(\tau,\mu_{1}^{\pm}), U(\tau,\mu_{1}^{\pm}), V(\tau,\mu_{1}^{\pm}), \ldots, I(\tau,\mu_{N/2}^{\pm}), Q(\tau,\mu_{N/2}^{\pm}), U(\tau,\mu_{N/2}^{\pm}), V(\tau,\mu_{N/2}^{\pm}) \right]^{T}, \tag{58}
$$

equation similar to Equation (22), can be derived (see [13,34]) but with all the matrices 4 times larger in each dimension.

To deal with the anisotropic part of the VRTE solution, we use the vector version of the MSH [13]:

$$
\overset{\rightarrow}{L}_{MSH}^{CP}(\tau,\nu,\psi) = \sum_{m=-2,0,2} \sum_{k=0}^{\infty} \frac{2k+1}{4\pi} \overset{\leftrightarrow}{Y}_{k}^{m}(\nu) \exp\left\{ -(\overset{\leftrightarrow}{1} - \Lambda\overset{\leftrightarrow}{x}_{k})\tau/\mu_{0} \right\} \exp(im\psi) \overset{\rightarrow}{f}_{k}^{m}(0), \tag{59}
$$

where $\overset{\leftrightarrow}{Y}_{m}^{k}(\mu) = \mathrm{diag}\begin{bmatrix} P_{m,+2}^{k}(\mu); & P_{m,+0}^{k}(\mu); & P_{m,-0}^{k}(\mu); & P_{m,-2}^{k}(\mu) \end{bmatrix}$.

## 12. Three-Dimensional Radiative Transfer Models

The analytical simplicity of solving a plane-parallel problem is determined by its symmetry. In the case of a three-dimensional medium (3D), such symmetry disappears and the BVP of the RTE in general cases must be solved numerically. In the numerical solution of the three-dimensional RTE, it is necessary to discretize the radiance field in the spatial and angular domains. To achieve the required accuracy of the RTE numerical solution, the fast convergence of the numerical calculation of the scattering integral over the total number of discrete angular directions is crucial. DOM is one of the gold standards for the angular discretization procedure [33]. Regardless of the specific scheme of the scattering integral representation for arbitrary 3D geometry of the medium, it is important to isolate the anisotropic part of the solution in order to reduce the number of discrete ordinates. As well as for the slab geometry, it is most appropriate to treat the anisotropic part of the solution by using the MSH, which has an analytical

form for any arbitrary geometry of the medium [13], thereby reducing the number of ordinates required for the scattering integral representation.

However, like in the slab geometry case, synthetic iterations can be applied here as well: we find the smooth part of the solution in the diffusion approximation, and then make use of iterations. This approach was referred to as "quasi-diffusion" when the direct radiation was isolated [35]. For spatially localized sources, MSH is accurate in the entire front hemisphere of the viewing directions; therefore, only one iteration from MSH is required in many practical problems.

A special case of arbitrary geometry is stochastic media, when the optical characteristics of the medium are distributed in space according to a certain probabilistic law, which leads to a stochastic uncertainty of the spatial-angular distribution of the radiance field. The transition to the equation for averaged quantities is complicated by the covariance terms, i.e., mean values of the products of the spatial-angular distribution of radiance and a random function of the optical parameters [36]. Subsequent averaging procedures lead to the hierarchical system of equations. To solve this system, certain closure conditions are assumed (e.g., neglecting higher-order covariance terms). However, such simplified stochastic models cannot reproduce several phenomena observed in nature, e.g., statistical lenses formed by the oceanic surfaces.

## 13. Inverse Models

To retrieve a given parameter of interest by using non-linear least-square fitting, the corresponding forward model must be linearized. Now, we consider the computations of the derivatives of the radiance with respect to parameter of interest, $\varsigma$. The analytical linearization method is one of the most efficient and generic techniques, as it can be applied to any implementation of the radiative transfer model. This approach is since the direct model is implemented as a sequence of derivatives of functions. The differential operator is applied, first, to the input parameters of the model. Then, by applying a chain rule, it propagates through the code until the derivatives of the radiances are computed. The linearization of the basic matrix operations is straightforward. The matrix multiplication is linearized by the Leibniz rule. The linearization of the eigenvalue problem (see Equation (26)) is not straightforward. As suggested in [37], we consider the eigenvalue problem with the normalization condition:

$$\begin{cases} BU = U\Gamma, \\ U^T U = 1. \end{cases} \tag{60}$$

Applying the differential operator to Equation (60) and using the orthogonality condition for eigenvectors, one can reach the following equation:

$$\begin{bmatrix} \mathbf{x} & \lambda\mathbf{E} - \mathbf{A} \\ 0 & \mathbf{x}^T \end{bmatrix} \begin{bmatrix} \frac{\partial\lambda}{\partial\varsigma} \\ \frac{\partial\mathbf{x}}{\partial\varsigma} \end{bmatrix} = \begin{bmatrix} \frac{\partial\mathbf{A}}{\partial\varsigma}\mathbf{x} \\ 0 \end{bmatrix}. \tag{61}$$

The derivatives for the initial eigenvalue problem for matrix B are expressed in terms of the solution of the system (61), which has to be solved for each layer and each azimuth harmonic. Later, by using the Leibniz rule, the derivatives for reflection and transmission matrices in the context of the matrix operator method (see Equations (29)–(31)) can be found.

## 14. Numerical Aspects

In practice, computing the anisotropic part of the radiance by using the small-angle approximation allows the number of discrete ordinates in the atmosphere to remain as low as two. For the ocean, at least four discrete ordinates are required. Note that the computations do not involve additional assumptions on the phase functions (i.e., no truncation procedures are required). The linearized model is just as accurate for the derivatives as the forward model for the radiances. Since all differentiations are performed "by hand", no additional assumptions are needed.

The coupled model MDOM has been compared with the coupled code LIDORT [37]. The results of the comparison are shown in Figure 2. The agreement up to the 6[th] digit has been obtained. The computations have been performed for the coupled model with the flat oceanic surface. The ocean has been modeled as a two-layer system with the dissolved organic matter in the upper oceanic layer. The atmosphere was modeled as a 14-layer system containing the trace gases ($O_3$, $NO_2$) as well as the urban aerosol.

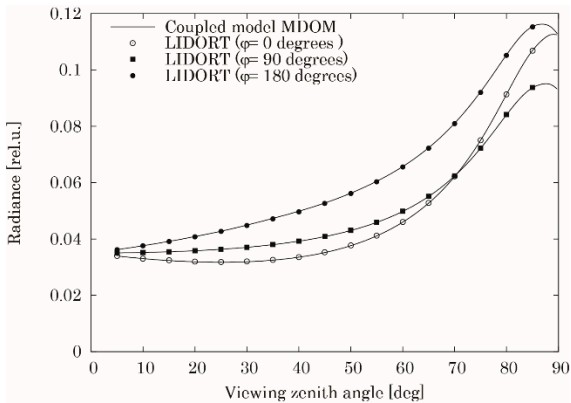

**Figure 2.** Validation of the coupled MDOM model against the coupled LIDORT model. The solar zenith angle is 35 degrees.

## 15. Conclusions

In this paper, our intension was to demonstrate modern advances in the discrete theory of radiative transfer. In a one-dimensional case, the theory provides an exact analytical solution as well as a numerical method for solving the radiative transfer equation. In this sense, a further comparison of different solvers reveals advantages of certain implementations (i.e., programming skills) or approximations rather than "new" methods behind the solvers [38]. Moreover, the comparison of various approximations looks misleading, because essentially, they are based on:

1) The synthetic iteration techniques equipped with the small-angle approximation, DOM or SHM or;
2) MOM with various degrees of assumptions.

The accuracy and limits of applicability of all these approximations can be estimated based on the discrete theory of radiative transfer. From the user perspective, an open-source reference tool for the preparation of atmospheric scenarios and benchmark cases would be very useful.

In our opinion, unsolved problems of radiative transfer are exclusively outside one-dimensional models. Three-dimensional [39] and stochastic radiative transfer models [8,40] for retrieval of atmospheric and water constituents are those "Thomson clouds" in the radiative transfer theory. However, they are at the interdisciplinary boundaries between radiative transfer and related fields, such as cloud physics and data processing [41,42].

**Author Contributions:** Conceptualization, V.P.B.; methodology, D.S.E.; software, A.Y.B.; validation, V.P.B., D.S.E. and A.Y.B.; formal analysis, A.A.K.; writing—original draft preparation, V.P.B.; writing—review and editing, D.S.E. and A.A.K.; supervision, V.P.A.; project administration, A.A.K. All authors have read and agreed to the published version of the manuscript.

**Funding:** This research received no external funding.

**Acknowledgments:** This paper is dedicated to our untimely departed friend, a true Knight of Ocean Optics, Leonid G. Sokoletsky.

**Conflicts of Interest:** The authors declare no conflict of interest.

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
