# Peer review of "Analysis of the Discrete Theory of Radiative Transfer in the Coupled “Ocean–Atmosphere” System: Current Status, Problems and Development Prospects"

_jmse, doi:10.3390/jmse8030202_

Round 1

Reviewer 1 Report

The revised work introduces more problems and needs revisions to clarify.

(1) The title is misleading. It covers a much wider area than what is actually presented in the manuscript. Also, the revised title reads like a review paper, which is in conflict with what says in the abstract as new analysis. Response to Reviewer 1, comment 3 is good, and should be reflected in the tilte change to refine the subject.

(2) The work does not use references properly. Eq. (1) shows up arbitrarily without crediting the original works. In fact, the whole process of analysis is built based on forward analysis of scattering based on matrix perturbation theorems shown in the Russian literature. The current work is brachned on top of proposed publications and should be extrememly clear about what is "borrowed" and what is "derived".  

(3) The complexity of the problem is expressed in a hand-waving style. The reivsed sentence "The inelastic scattering processes changing the wavelength of the light are not included." Why is the neglection justified? Is it because it does not exist or accounts for 30% of the scattering event? The reasons must be presented instead of a gesture of "throwing away".

(4) What do the authors mean with the added sentence " boundary value problem (1) is a set of equations."? What is the "set of questions?

(5) One obvious problem for Eq. (10) is that in using Ref[9], the outcome doesn't not integrate well with imposed analysis of L2 in Eq. (9) and the set of solutions from Eq. (7), which is belived to be the authors new anlaysis. At least one of the terms should be expanded/explained in terms of the aforementioned analysis, and the rest be pointed to anlaysis that is not included in the current work.

(6) With Ambartsumian's work included, plus refs [21-24]. It is clear that the latter half of the work is not new and should be clarified as a review paper.

(7) I remain skeptical for MDOM as its claims sound false: "MDOM provides average convergence" define "average convergence" (non-scientific language). "All methods for selecting the anisotropic part are equivalent to each other in a uniform metric" define all methods? Personally I believe it should be [a uniform metric is used to select the anisotropic part]. The original statement sounds more complex than it needs to be. "In order to achieve good convergence in a uniform metric, the sampling interval must
correspond to the angular size of the finest detail of the radiance distribution to be reproduced" vague claim. The aim for "finest detail" is to minimize global truncation error in the forward "Eular method" like numerical analysis. Convergence doesn't exclude systematic bias.

(8) Revised lines 350-353 is confusing. Especially "... by the possibility of the phase function to be expanded in series" what does it mean ? Please reword these sentences.

(9) Added sentence "thereby reducing the number of ordinates required for the scattering integral representation." Then what? Is it "reducing the complexity of 3D calcualtion at the cost of certain accuracies"? The section simply suggest application to 3D is non-conclusive and potential methods are suggested to make the problem tractable. The current writing is unclear in meaning.

(10) Conclusion "our intension was to demonstrate the completeness of the discrete theory of radiative transfer achieved in more than 100 years." is a strange claim. It is practical to point out what is achieved other than "intentions". A lot of complexity in RT is not included in the disucssion, such as turbulence, general aerosol effects and etc. Claiming "completeness" is problematic. The rest of the conclusion should be clear about what is solved and what is future work.

In review of the authors response in the previous review cycle, it doesn't seem that the authors acknowledge the complexity of the RT problem, but rather avoids facts and evidences that are not accounted for in the work of their analysis. Remember that solving a fraction of the RT problem is as good as solving its many other aspects. Admitting factors not included in current modelings doesn't hurt the true value of this work, and the authors are encouraged to refine their work with a well-focused aspect.    

Author Response

Point 1: The title is misleading. It covers a much wider area than what is actually presented in the manuscript. Also, the revised title reads like a review paper, which is in conflict with what says in the abstract as new analysis. Response to Reviewer 1, comment 3 is good, and should be reflected in the title change to refine the subject.

Response 1: Unfortunately, we are not quite following the reviewers’ comments at this point. The essence of the title refers to the citation of the classical textbook of S.Chandrasekhar “Radiative transfer” which is about radiation propagation considering the light as rays. In Abstract we refer to the same problem, as in title, namely, we are talking about DISCRETE radiative transfer. We would be grateful if the reviewer could precisely point out the contradiction in our paper.

Point 2: The work does not use references properly. Eq. (1) shows up arbitrarily without crediting the original works. In fact, the whole process of analysis is built based on forward analysis of scattering based on matrix perturbation theorems shown in the Russian literature. The current work is branched on top of proposed publications and should be extrememly clear about what is "borrowed" and what is "derived".

Response 2: The RTE is the classic equation of mathematical physics. In this regard, it seems to us that the references here are redundant, just as they are not necessary when referring to the works of Maxwell, Newton, Boltzmann etc. The purpose of our work is to analyze methods for solving RTE boundary value problems of the atmosphere-ocean system, wherein numerical comparisons and scientific disputes have not subsided so far. We note that we have cited only 2 papers which are in Russian. The essence of the work is the analysis of solution methods. Their internal unity is shown in the framework of the discrete theory of transfer. Based on the performed analysis we highlighted the urgent tasks of the theory of transfer today.

Point 3: The complexity of the problem is expressed in a hand-waving style. The reivsed sentence "The inelastic scattering processes changing the wavelength of the light are not included." Why is the neglection justified? Is it because it does not exist or accounts for 30% of the scattering event? The reasons must be presented instead of a gesture of "throwing away".

Response 3: The purpose of the article is not to solve a specific problem of radiation transfer in a real atmosphere-ocean system, but to analyze methods for solving UPI and determine the actual problems of the theory. In the Introduction, paragraphs are added (highlighted in yellow) that accurately define the model of the medium and the essence of the discrete theory.

Speaking about inelastic scattering, we would like to mention the following. We are confused by the comment about 30 %. The ratio between inelastic and elastic cross-sections is 4% in the visible spectral range: Wagner T., Beirle S., Deutschmann T. Three-dimensional simulation of the Ring effect in observations of scattered sun light using Monte Carlo radiative transfer models. Atmospheric Measurement Techniques, 2009, 2, 1, 113–124.

Point 4: What do the authors mean with the added sentence " boundary value problem (1) is a set of equations."? What is the "set of questions?

Response 4: Corrections were made to the text of the article.

Point 5: One obvious problem for Eq. (10) is that in using Ref[9], the outcome doesn't not integrate well with imposed analysis of L2 in Eq. (9) and the set of solutions from Eq. (7), which is believed to be the authors new analysis. At least one of the terms should be expanded/explained in terms of the aforementioned analysis, and the rest be pointed to analysis that is not included in the current work.

Response 5: We explained the essence of the solution in the paper.

Point 6: With Ambartsumian's work included, plus refs [21-24]. It is clear that the latter half of the work is not new and should be clarified as a review paper.

Response 6: In section 7, we showed that the discrete matrix solution is reduced to the same results that were obtained analytically by Ambartsumian. It is also important to emphasize that the property of invariance is contained in the matrix solution. In our opinion, these are new results. If we are wrong, we kindly ask the reviewer to explicitly indicate a reference where the invariance is formulated in the matrix form, and the results of Ambartsumian are obtained from it.

Point 7: I remain skeptical for MDOM as its claims sound false: "MDOM provides average convergence" define "average convergence" (non-scientific language). "All methods for selecting the anisotropic part are equivalent to each other in a uniform metric" define all methods? Personally, I believe it should be [a uniform metric is used to select the anisotropic part]. The original statement sounds more complex than it needs to be. "In order to achieve good convergence in a uniform metric, the sampling interval must correspond to the angular size of the finest detail of the radiance distribution to be reproduced" vague claim. The aim for "finest detail" is to minimize global truncation error in the forward "Eular method" like numerical analysis. Convergence doesn't exclude systematic bias.

Response 7: It was an incorrect translation error, replaced with "convergence in mean" [http://mathworld.wolfram.com/ConvergenceinMean.html, Swartz C. Measure integration and function spaces, 1994]. The text has been changed

Point 8: Revised lines 350-353 is confusing. Especially "... by the possibility of the phase function to be expanded in series" what does it mean? Please reword these sentences.

Response 8: The text has been amended accordingly.

Point 9: Added sentence "thereby reducing the number of ordinates required for the scattering integral representation." Then what? Is it "reducing the complexity of 3D calcualtion at the cost of certain accuracies"? The section simply suggest application to 3D is non-conclusive and potential methods are suggested to make the problem tractable. The current writing is unclear in meaning.

Response 9: It is one of our results – here is one of the unsolved problems, namely, 3D radiative transfer. It seems to us that it is possible to apply the techniques described in the one-dimensional case to the 3D case.

Point 10: Conclusion "our intension was to demonstrate the completeness of the discrete theory of radiative transfer achieved in more than 100 years." is a strange claim. It is practical to point out what is achieved other than "intentions". A lot of complexity in RT is not included in the discussion, such as turbulence, general aerosol effects and etc. Claiming "completeness" is problematic. The rest of the conclusion should be clear about what is solved and what is future work.

Response 10: We have changed the text of this sentence.

The main scope of the discrete radiative transfer was formulated in sections 1 and 2. All effects related to turbulence and general aerosols are out of the scope of the discrete theory since the optical properties of the system "atmosphere-ocean" are assumed to be known (or explicitly given).

Point 11: In review of the authors response in the previous review cycle, it doesn't seem that the authors acknowledge the complexity of the RT problem, but rather avoids facts and evidences that are not accounted for in the work of their analysis. Remember that solving a fraction of the RT problem is as good as solving its many other aspects. Admitting factors not included in current modelling doesn't hurt the true value of this work, and the authors are encouraged to refine their work with a well-focused aspect.

Response 11: It seems to us that the reviewer considers the problem of transfer theory more broadly than we see it. For us, the transfer theory is a consequence of the ray, photometric model of light representation [1, 2]. Anything that goes beyond that goes beyond our work. When we talk about the completeness, we do not pretend to study problems that go beyond ray representations.

Reviewer 2 Report

I have no additional comments on the revised paper.  

Author Response

Point 1: In the introduction Authors should clearly state what they mean by the term “Discrete Theory of Radiate Transfer”. Indeed, a significant part of the numerical methods is based on the use of the procedure of discretization of equations, boundary conditions, relations, etc. This kind of term is more appropriate to use when studying the process of radiation transfer in densely packed media.

Response 1: We added a paragraph in the Introduction, where we explained our understanding of the term "discrete theory" - marked in yellow.

Point 2: In the introduction of the manuscript the authors should briefly list achievements of the rigorous theory of radiation transfer and their usefulness for evaluating the accuracy of solutions (BVPs for RTE) obtained using various approximate methods. Authors should more objectively analyze the development of RTT over a period of 100 years. In particular, the manuscript contains no mention of the works by Chandrasekhar and Wick (1943-1950). It was these scientists who were the first to use discrete ordinates and spherical harmonics methods to solve RTT problems. There is no reference in the manuscript to the classic monograph by Case and Zweifel (1967). It contains in sufficient detail the information related to the definition and role of Green's functions in RTT, as well as an analysis of discrete ordinates and spherical harmonics methods.

Response 2: Analytical solutions are outside the scope of this article. We consider it redundant to give references to the classics of the transfer theory. Do not cite references to Newton, Maxwell, Boltzmann, ... We assume that readers unfamiliar with Chandrasekhar, Wick, Eddington, Milne, Jeans, ... will not be. Our article focuses on the analysis of the development of numerical methods for solving RTE and determining its existing problems. References to the works of Case, Zweifel, and Chandrasekar are given in the text.

Point 3: Line 259 contains the phrase “applying the spherical harmonics method [21]”. The co-author of [21] is V.P. Budak, who is the co-author of the manuscript. But V.P. Budak is not the founder of the spherical harmonics method. Besides link [21] the authors should add references to the works of Wick (1943) and Case, Zweifel (1967). Line 115 contains the sentence "In [7], it was shown that any boundary value problem of radiative transfer can be derived from the problem with the point unidirectional source". D.S. Efremenko is one of the authors of the manuscript and article [7]. But the statement made in line 115 has been known for a long time (see, Case's and Zweifel's monograph). Therefore, a reference to this monograph should be added to the references [7, 10]. The old and new versions of the manuscript contain many examples of biased citation of publications. The authors did not provide references to works in which rigorous, asymptotic, and high-precision numerical solutions of various BVPs were obtained (including multidimensional problems). Moreover, a number of these solutions were obtained for the case of arbitrary phase functions. Since these works were published for the period from 2010 to 2017 in well-known and accessible scientific publications (in particular, in LSR-5, JQSRT, Astrophysics and Space Sciences, DE, ...), in reviewer's judgment, the authors either consciously do not refer to these works, or do not fully understand the value of accurate solutions (and their rigorous evaluations) for testing the approach developed by them to solving RTT problems. It should be noted that a real accuracy of any approximate solutions can be made only when there are test (exact) solutions.

Response 3: The text has been corrected to eliminate errors and ambiguity of perception.

Point 4: The misprints in Eqs. (6), (14), (45) should be corrected in the new version of the manuscript.

Response 4: Unfortunately, we don't see any misprints in these equations. We would be very grateful if the reviewer pointed out them explicitly. Thank you in advance.

Point 5: In lines 321,322,333-335,340,341,379-381 etc. the authors in fact categorically formulated a number of statements without any justification. Moreover, a part of these statements can only be partially valid within the framework of certain limitations. Authors should give references to papers that substantiate these claims or soften the formulation of these statements.

Response 5: The text contained in the old version of the manuscript in lines 321, 322, 333-335 has been changed. All work is written on a physical level of rigor. In 340, 341, we do not see any misunderstandings – the results of comparisons are given [23]. For 379-381, the phrase "in general cases" was added.

Point 6: The manuscript does not contain specific initial data, within which the solutions obtained on the basis of various algorithms were compared (see, Fig. (3)). Line 433 contains the statement “The agreement up to the 6th digit has been obtained”, which does not have a universal meaning as a comparison was made between the results obtained by approximate methods. There would be much more confidence in their results if the authors compared (within the framework of a simpler model) the results of their calculations with exact solutions.

Response 6: A comparison is made with an exact analytical solution for a semi-infinite medium with Rayleigh scattering.

Point 7: I will now note a number of mathematical facts. They will allow the authors to understand better the essence of the above statements and comments of the reviewer. These facts should not be ignored in estimating the errors of solutions obtained using approximate methods. However, the contents of some parts of the manuscript text indicates that the authors ignore them. According to the qualitative mathematical theory of boundary value problems for RTE, the features of their solutions (and their derivatives) can arise for several reasons. These features appear due to the presence of boundaries between regions that have different geometric and physical (in particular, optical) properties. In addition, such features arise if the local transmission and reflection operators on different sides of the boundaries themselves have singularities in spatial and angular variables. Such a situation takes place, for example, at a sea-atmosphere interface. Features of BVPs of the RTE solutions also appear when the functions describing the external and internal radiation sources and the functions that determine the laws of radiation scattering by the elements of the medium themselves have features. In turn, this type of feature leads to negative consequences when using representations of solutions of BVPs and phase functions in the form of sums or series in orthogonal systems of functions. In particular, Eqs. (15), (34), (45), (46). (47), (51), (53) are examples of such expansions in the manuscript. As solutions of BVPs for the RTE (in particular, on the boundaries of media) are nondifferentiable functions with respect to spatial and angular variables on closed sets (for example, on the intervals [-1,1], [0,1], [- 1,0] and etc), a number of non-trivial problems related to the correct estimation of the accuracy of approximations of these solutions arise. I will give for illustration only the Faber theorem, which, in essence, limits the universality and accuracy of the statements and procedures presented in the manuscript. Faber's theorem. Whatever the strategy of choosing the interpolation nodes, there exists a function y = f (x) continuous on the segment [a, b], for which the following relation held:

max|f(x)-P(x;n)|→+∞,n→+∞,

x∈[a.b]

where P (x; n) is the nth order interpolation polynomial. It should be noted that if the function y = f (x) is differentiable, then such a strategy exists, for example, for a system of Chebyshev polenomias. However, in the framework of the turbid media model considered by the authors, the solutions of BVPs for the RTE are not, generally speaking, differentiable (in particular, at a sea-atmosphere interface).The well-known error estimates of the Gaussian quadrature formulas used by the authors are expressed in terms of the upper estimates of the moduli of derivatives with respect to angular variables on closed sets. The situation becomes even more complicated for the case of multi-dimensional integrals. The reviewer knows that ignoring or not enough accurately accounting for the non-differentiability of BVPs for the RTE solutions, even in fairly small areas (in angular variables), can lead to noticeable deviations of approximate solutions from true (test) solutions. Therefore the authors of the manuscript should not affirm that solutions of BVPs for the RTE can be obtained with high accuracy using the procedure described in the manuscript to minimize the number of nodes used. Minimizing the time to obtain numerical solutions with high accuracy is not always compatible with this accuracy.

Response 7: This problem exists and is described in detail in the monograph Case, Zweifel. However, in MDOM, it is strongly mitigated by two techniques. First, subtracting the MSH, which makes the boundary conditions null. Second, the Sykes scheme is used, which separates the streams up and down. The correct choice of the number of discrete ordinates and the use of synthetic iterations is also essential.

Reviewer 3 Report

The Second review to the manuscript

«Discrete Theory of Radiation Transfer in The Coupled «Ocean-Atmosphere System: Current Status, Problems, Development Prospects»

The reviewer after analyzing the contents of the new text of manuscript came to the conclusion that this work can be published in the Journal of Marine Science and Engineering only after clarifying the terminology, correcting incorrect statements and expanding the list of cited literature. In addition, it would be very useful to increase the number of plots, to add tabular data that would illustrate the effectiveness of approximate methods for solving radiative transfer theory (RTT) problems described in the manuscript. The reviewer was forced to draw such a conclusion because the authors of the manuscript essentially ignored a number of  recommendations that he gave in his first review. Authors should modify the text of the manuscript so that its contents would be interesting not only to potential readers with a tangential knowledge of the basics of RTT, but would be of interest to qualified specialists in this theory and its applications as well.Now I will give a list of remarks and explanations that the authors should take into account when finalizing the contents of the manuscript text.1. In the introduction Authors should clearly state what they mean by the term “Discrete Theory of Radiate Transfer”. Indeed, a significant part of the numerical methods is based on the use of the procedure of discretization of equations, boundary conditions, relations, etc. This kind of term is more appropriate to use when studying the process of radiation transfer in densely packed media.2. In the introduction of the manuscript the authors should briefly list achievements of the rigorous theory of radiation transfer and their usefulness for evaluating the accuracy of solutions (BVPs for RTE) obtained using various approximate methods. Authors should more objectively analyze the development of RTT over a period of 100 years. In particular, the manuscript contains no mention of the works by Chandrasekhar and Wick (1943-1950). It was these scientists who were the first to use discrete ordinates and spherical harmonics methods to solve RTT problems.   There is no reference in the manuscript to the classic monograph by Case and Zweifel (1967). It contains in sufficient detail the information related to the definition and role of Green's functions in RTT, as well as an analysis of discrete ordinates and spherical harmonics methods.

  1. Line 259 contains the phrase “applying the spherical harmonics method [21]”. The co-author of [21] is V.P. Budak, who is the co-author of the manuscript. But V.P. Budak is not the founder of the spherical harmonics method. Besides link [21] the authors should add references to the works of Wick (1943) and Case, Zweifel (1967).Line 115 contains the sentence "In [7], it was shown that any boundary value problem of radiative transfer can be derived from the problem with the point unidirectional source". D.S. Efremenko is one of the authors of the manuscript and article [7]. But the statement made in line 115 has been known for a long time (see, Case's and Zweifel's monograph). Therefore, a reference to this monograph should be added to the references [7, 10]. The old and new versions of the manuscript contain many examples of biased citation of publications.The authors did not provide references to works in which rigorous, asymptotic, and high-precision numerical solutions of various BVPs were obtained (including multidimensional problems). Moreover, a number of these solutions were obtained for the case of arbitrary phase functions. Since these works were published for the period from 2010 to 2017 in well-known and accessible scientific publications (in particular, in LSR-5, JQSRT, Astrophysics and Space Sciences, DE, ...), in reviewer's judgment, the authors either consciously do not refer to these works, or do not fully understand the value of accurate solutions (and their rigorous evaluations) for testing the approach developed by them to solving RTT problems.It should be noted that a real accuracy of any approximate solutions can be made only when there are test (exact) solutions.4. The misprints in Eqs. (6), (14), (45) should be corrected in the new version of the manuscript.5. In lines 321,322,333-335,340,341,379-381 etc. the authors in fact categorically formulated a number of statements without any justification. Moreover, a part of these statements can only be partially valid within the framework of certain limitations. Authors should give references to papers that substantiate these claims or soften the formulation of these statements.6. The manuscript does not contain specific initial data, within which the solutions obtained on the basis of various algorithms were compared (see, Fig. (3)).Line 433 contains the statement “The agreement up to the 6th digit has been obtained”, which does not have a universal meaning as a comparison was made between the results obtained by approximate methods. There would be much more confidence in their results if the authors compared (within the framework of a simpler model) the results of their calculations with exact solutions.

         I will now note a number of mathematical facts. They will allow the authors to understand better the essence of the above statements and comments of the reviewer. These facts should not be ignored in estimating the errors of solutions obtained using approximate methods. However, the contents of some parts of the manuscript text indicates that the authors ignore them.According to the qualitative mathematical theory of boundary value problems for RTE, the features of their solutions (and their derivatives) can arise for several reasons. These features appear due to the presence of boundaries between regions that have different geometric and physical (in particular, optical) properties. In addition, such features arise if the local transmission and reflection operators on different sides of the boundaries themselves have singularities in spatial and angular variables. Such a situation takes place, for example, at a sea-atmosphere interface. Features of BVPs of the RTE solutions also appear when the functions describing the external and internal radiation sources and the functions that determine the laws of radiation scattering by the elements of the medium themselves have features. In turn, this type of feature leads to negative consequences when using representations of solutions of BVPs and phase functions in the form of sums or series in orthogonal systems of functions. In particular, Eqs. (15), (34), (45), (46). (47), (51), (53) are examples of such expansions in the manuscript.As solutions of BVPs for the RTE (in particular, on the boundaries of media) are nondifferentiable functions with respect to spatial and angular variables on closed sets (for example, on the intervals [-1,1], [0,1], [- 1,0] and etc), a number of non-trivial problems related to the correct estimation of the accuracy of approximations of these solutions arise.I will give for illustration only the Faber theorem, which, in essence, limits the universality and accuracy of the statements and procedures presented in the manuscript.Faber's theorem. Whatever the strategy of choosing the interpolation nodes, there exists a function y = f (x) continuous on the segment [a, b], for which the following relation held:

max|f(x)-P(x;n)|→+∞,n→+∞,

x∈[a.b]

where P (x; n) is the nth order interpolation polynomial.It should be noted that if the function y = f (x) is differentiable, then such a strategy exists, for example, for a system of Chebyshev polenomias.However, in the framework of the turbid media model considered by the authors, the solutions of BVPs for the RTE are not, generally speaking, differentiable (in particular, at a sea-atmosphere interface).The well-known error estimates of the Gaussian quadrature formulas used by the authors are expressed in terms of the upper estimates of the moduli of derivatives with respect to angular variables on closed sets. The situation becomes even more complicated for the case of multi-dimensional integrals. The reviewer knows that ignoring or not enough accurately accounting for the non-differentiability of BVPs for the RTE solutions, even in fairly small areas (in angular variables), can lead to noticeable deviations of approximate solutions from true (test) solutions.Therefore the authors of the manuscript should not affirm that solutions of BVPs for the RTE can be obtained with high accuracy using the procedure described in the manuscript to minimize the number of nodes used. Minimizing the time to obtain numerical solutions with high accuracy is not always compatible with this accuracy. 

Author Response

Point 3: Line 259 contains the phrase “applying the spherical harmonics method [21]”. The co-author of [21] is V.P. Budak, who is the co-author of the manuscript. But V.P. Budak is not the founder of the spherical harmonics method. Besides link [21] the authors should add references to the works of Wick (1943) and Case, Zweifel (1967). Line 115 contains the sentence "In [7], it was shown that any boundary value problem of radiative transfer can be derived from the problem with the point unidirectional source". D.S. Efremenko is one of the authors of the manuscript and article [7]. But the statement made in line 115 has been known for a long time (see, Case's and Zweifel's monograph). Therefore, a reference to this monograph should be added to the references [7, 10]. The old and new versions of the manuscript contain many examples of biased citation of publications. The authors did not provide references to works in which rigorous, asymptotic, and high-precision numerical solutions of various BVPs were obtained (including multidimensional problems). Moreover, a number of these solutions were obtained for the case of arbitrary phase functions. Since these works were published for the period from 2010 to 2017 in well-known and accessible scientific publications (in particular, in LSR-5, JQSRT, Astrophysics and Space Sciences, DE, ...), in reviewer's judgment, the authors either consciously do not refer to these works, or do not fully understand the value of accurate solutions (and their rigorous evaluations) for testing the approach developed by them to solving RTT problems. It should be noted that a real accuracy of any approximate solutions can be made only when there are test (exact) solutions.

Response 3: The text has been corrected to eliminate errors and ambiguity of perception.

Point 7: I will now note a number of mathematical facts. They will allow the authors to understand better the essence of the above statements and comments of the reviewer. These facts should not be ignored in estimating the errors of solutions obtained using approximate methods. However, the contents of some parts of the manuscript text indicates that the authors ignore them. According to the qualitative mathematical theory of boundary value problems for RTE, the features of their solutions (and their derivatives) can arise for several reasons. These features appear due to the presence of boundaries between regions that have different geometric and physical (in particular, optical) properties. In addition, such features arise if the local transmission and reflection operators on different sides of the boundaries themselves have singularities in spatial and angular variables. Such a situation takes place, for example, at a sea-atmosphere interface. Features of BVPs of the RTE solutions also appear when the functions describing the external and internal radiation sources and the functions that determine the laws of radiation scattering by the elements of the medium themselves have features. In turn, this type of feature leads to negative consequences when using representations of solutions of BVPs and phase functions in the form of sums or series in orthogonal systems of functions. In particular, Eqs. (15), (34), (45), (46). (47), (51), (53) are examples of such expansions in the manuscript. As solutions of BVPs for the RTE (in particular, on the boundaries of media) are nondifferentiable functions with respect to spatial and angular variables on closed sets (for example, on the intervals [-1,1], [0,1], [- 1,0] and etc), a number of non-trivial problems related to the correct estimation of the accuracy of approximations of these solutions arise. I will give for illustration only the Faber theorem, which, in essence, limits the universality and accuracy of the statements and procedures presented in the manuscript. Faber's theorem. Whatever the strategy of choosing the interpolation nodes, there exists a function y = f (x) continuous on the segment [a, b], for which the following relation held:

max|f(x)-P(x;n)|→+∞,n→+∞,

x∈[a.b]

where P (x; n) is the nth order interpolation polynomial. It should be noted that if the function y = f (x) is differentiable, then such a strategy exists, for example, for a system of Chebyshev polenomias. However, in the framework of the turbid media model considered by the authors, the solutions of BVPs for the RTE are not, generally speaking, differentiable (in particular, at a sea-atmosphere interface).The well-known error estimates of the Gaussian quadrature formulas used by the authors are expressed in terms of the upper estimates of the moduli of derivatives with respect to angular variables on closed sets. The situation becomes even more complicated for the case of multi-dimensional integrals. The reviewer knows that ignoring or not enough accurately accounting for the non-differentiability of BVPs for the RTE solutions, even in fairly small areas (in angular variables), can lead to noticeable deviations of approximate solutions from true (test) solutions. Therefore the authors of the manuscript should not affirm that solutions of BVPs for the RTE can be obtained with high accuracy using the procedure described in the manuscript to minimize the number of nodes used. Minimizing the time to obtain numerical solutions with high accuracy is not always compatible with this accuracy.

Response 7: This problem exists and is described in detail in the monograph Case, Zweifel. However, in MDOM, it is strongly mitigated by two techniques. First, subtracting the MSH, which makes the boundary conditions null. Second, the Sykes scheme is used, which separates the streams up and down. The correct choice of the number of discrete ordinates and the use of synthetic iterations is also essential.

Round 2

Reviewer 1 Report

I have read the response and revised paper. Comments 1,3, 11 are not answered properly. When you read the reply for 1 and 3, you will find that the authors are contradicting themselves. I recall earlier that I commented to you by saying that the only major problem is in comment #1 where the authors are overclaiming what they are doing without proper justification. The rest are basically minor issues. I do not think this is realistic that "radiation transfer" means rays in their response to comment #1. The authors have also added material that is not recommended in the previous review, such as the fuzzy figure 2. I believe that the journal's review process should not be used for iterative progress, and my recommendation stays as rejection.

Author Response

Point 1: Comments 1,3, 11 are not answered properly. When you read the reply for 1 and 3, you will find that the authors are contradicting themselves.

Response 1: We do not see any contradictions in our answers. Transfer theory is a consequence of the ray (photometric) approximation. The article does not analyze the solution to any specific practical problem. Therefore, we do not explain the conditions for the applicability of transfer theory. We analyze the current state of the discrete theory of radiative transfer, formulate its actual problems and possible methods for solving them. We will be grateful to the reviewer if he indicates specific contradictions to us.

Point 2: I recall earlier that I commented to you by saying that the only major problem is in comment #1 where the authors are overclaiming what they are doing without proper justification.

Response 2: We analyze the current state of numerical methods for solving the radiative transfer equation. Wherever we use other author's results, we provide references. If this is not so, then we will be grateful to the reviewer if he points out specific examples of using other author's results without references to the original works.

Point 3: The rest are basically minor issues. I do not think this is realistic that "radiation transfer" means rays in their response to comment #1.

Response 3: We not only assert that radiative transfer is a consequence of the ray approximation but also give two new references [1,2] where this is strictly shown. The first of them is the classic recognized monograph of Apresyan-Kravtsov. The second link is our article, where we consistently and rigorously show how the transfer equation follows from the Maxwell equations in the ray approximation using the Keldysh formalism. The list of works can be substantially increased, but this will already go beyond the scope of our work.

Let us only give one example of an excellent review of the works in this area: Mishchenko, M.I. Directional radiometry and radiative transfer: A new paradigm. Journal of Quantitative Spectroscopy & Radiative Transfer, 2011, 112, 2079–2094

Point 4: The authors have also added material that is not recommended in the previous review, such as the fuzzy figure 2.

Response 4: Figure 2 gives an estimate of the accuracy of the complete solution MDOM based on a comparison of calculations of the radiance of the reflected radiation with a well-known and widely used program LIDORT. The agreement up to the 6th digit has been obtained. This figure seems to us to be one of the most important parts of the proposed article.

Point 5: I believe that the journal's review process should not be used for iterative progress, and my recommendation stays as rejection.

Response 5: In our opinion, in response to the comments of reviewers, we only explained our results. Nowhere, we have not changed the formulation of the scientific problem, the results obtained, and the methodology for their justification.

This manuscript is a resubmission of an earlier submission. The following is a list of the peer review reports and author responses from that submission.

Round 1

Reviewer 1 Report

The manuscript “Discrete Theory of Radiation Transfer in The Coupled “Ocean-Atmosphere” System: Current Status, Problems, Development Prospects” provides the review of a simplified analysis of radio wave transmission from ocean to atmosphere where the media is viewed as layered slabs with unified refractive property. It is important to note that the title is not accurate, as radiation transfer covers a wide spectra including optical wavelengths that will cause complication to the asserted theory. To use such title without over-claims, the authors should provide additional statements on the targeted wavelength for such analysis, as well as complications if the model were to be applied to other wavelengths. For additional information, optical transmission through atmospheric turbulence is a hard problem and is often combined with aerosol extinctions, the thinking in the authors work is usable but doesn’t necessarily meet the uniformity assumption in propagation. The authors are welcome to check the literature and a fairly recent experimental work “Multi-aperture laser transmissometer system for long-path aerosol extinction rate measurement," Appl. Opt. 57, 551-559 (2018).

The reviewer does not see sufficient references listed when stating the equations, which makes it arbitrary to see Eq. (1) and so forth. To make the reading pleasant, the authors should summarize before the modeling equations on A. List of assumptions B. List of symbolic representations. For example, for A, the vector and matrix representation of the transmission and reflection parts should be stated to clarify the dimensionalized matrix approximation. For B, it is better to state the symbols \tau, I_hat, and so on beforehand with concise interpretation of their physics meanings to help readers go through the manuscript. The reviewer often find it hard to interpret the intended meanings without going back and forth.

Line 75-81: when defining the haze terms and modeling it, it is better to clarify the assumptions on transmission where the alleged resolution only sees the binary result of either \delta or 0 for radiation transmission through a “haze” grid point. While lambertion scattered signal is not affected by the directional block of “haze”.

Line 84-91: for the linear perturbation theory, say that \rou_bar is the ensemble average of the scattering area and then list its definition equation. Change equation (6) to something like <||\rou_tilda||>  <<  \rou_bar, as the integral itself is zero by its original definition.

The reviewer doesn’t see discussion regarding absorption. Is it reasonable to say that the equations are built based on relative scales when absorption loss is compensated to keep the conservative law of energy per scattering and propagation step? And the overall effect can be tuned with a fudge-factor?

Section 5, when discretize the result, the wavelength dependency is not reflected. Am I right to understand that the transmission analysis is a single wavelength and its wavelength-dependency parameterized in the values of transmission matrices?

Anisotropic effects are common observations of true radiation transmittance through turbid media, the current manuscript is not clear regarding how anisotropy is considered in the modeling works. Part 8 is a little confusing and maybe more reference to others’ works should be added here.

Part 11-14 can be viewed as model variations that are not exactly in line with the main thread of stating the method. I suggest including them into a single section “model variations” and using sub-titles to state individual issues.

Overall, I believe major revision is required for the manuscript.

Reviewer 2 Report

This paper is a very theoretical and mathematical treatment of certain unifying ideas in radiative transfer theory.  As such, it will be of interest to a small (but important) community of researchers.  I therefore recommend it for publication, with the understanding that the paper will not be of general interest to people who use radiative transfer theory numerical models for solving practical problems in atmospheric or oceanic optics.

I have only a few minor comments on the writing:

The title:  “Discrete Theory…:  Current Status, Problems, Development Prospects”  This led me to expect that there would be a review of problems with currently available radiative transfer models (e.g. missing physics or inaccurate numerical solution techniques), and how those problems could be corrected.  I don’t see anything like that in the paper.  It is a formal development of radiative transfer theory, centered mainly on the Discrete Ordinates Method without any discussion of “Problems or Development Prospects.”

Lines 28-37.  The comments on the status of physics in the late 1800s are correct.  However, I don’t find anything in the paper that corresponds to the “clouds” mentioned by Thompson in his 1901 paper.  Either the “clouds” in current radiative transfer theory need to be explicitly named and discussed, or this paragraph should be omitted.  Also, I would say that Thompsons “Cloud (a)” (speed of light) led to special relativity, and that his “Cloud (b)” (blackbody radiation) led to quantum mechanics.

Lines 52-53:  I don’t know what “stochastic lenses that allow seeing the bottom of the ocean under certain circumstances” refers to.  This needs more explanation.  (I guess this is one of the “clouds” in radiative transfer theory mentioned previously, but there is no further discussion of this.)

Lines 58, 60, and other places.  The authors use a comma to denote the vector inner product, e.g., (i,I) in line 58 and (z,I) in line 60.  This is OK, but at least in the US, a centered dot is more common, e.g. i.I in line 58 and z.I in line 60.

Line 73 and 78:  “the ‘haze’ component”.  I would call this “the diffuse component”

Line 99: “a reflecting Lambert bottom” should be “a reflecting Lambertian bottom”

Line 181: “…the matrices R ? T are the…”  In my document, the “?” looks like the Cyrillic letter “i” (a backward N).  I think this should be “…the matrices R and T are the…”

Line 437:  “inter-comparison”  I would just say “comparison”.  I have never understood what an “intercomparison” is, compared to a “comparison”.

Reviewer 3 Report

Review  to the manuscript

«Discrete Theory of Radiation Transfer in The Coupled «Ocean-Atmosphere System: Current Status, Problems, Development Prospects»

The reviewer after studying the contents of the text of this manuscript came to the conclusion that this work can be published in the Journal of Marine Science and Engineering only after clarifying the terminology, correcting incorrect statements and expanding the list of cited literature. In addition, it would be very useful to increase the number of plots, to add tabular data that would illustrate the effectiveness of approximate methods for solving radiative transfer theory (RTT) problems described in the manuscript.

     Now I will give a list of remarks that the authors should take into account when finalizing the contents of the manuscript text.

Authors should clarify the meaning of the statements in Abstract and remove sentences from it that do not correspond to the essence of the material presented in the work. The first and third sentences of Abstract are such statements. Indeed, in the manuscript only the part of the RTT that relates to the development and application of approximate semi-analytical or individual numerical methods is analyzed. In Introduction (see lines 35.36) there is a phrase «Our goal in this paper is to analyze the current state of the radiative transfer theory and find its main problems, i.e. …». However, the authors consider in the manuscript only approximate methods for solving BVPs for RTE. They do not provide references to works in  which  rigorous methods for solving BVPs for RTE are presented or highly accurate and tested solutions to a number of such problems are obtained for cases when it is necessary to use several thousand terms in their expansion in Fourier series in Legendre polynomials for the correct approximation of phase functions.  For example, to correctly describe the phase functions corresponding to the optical properties of sea water, it is necessary to use about 2000 or more members in their expansions in Legendre polynomials. In the works of the authors to which they refer, the number of such members was no more than 801. To date, papers have already been published in which not only rigorous analytical expressions for the reflection function, plane and spherical albedos are obtained, but also high-precision and tested numerical values for these quantities are found for situations where phase functions are sharply anisotropic within the unit sphere. At the same time, for the correct approximation of such phase functions, it was necessary to use about 2100, 3800 and 5400 members in their expansion into Fourier series in Legendre polynomials. In particular, the following publications are such works: a)Rogovtsov, N.N.; Borovik,F. Application of general invariance relations reduction method to solution of radiation transfer problems. J. Quant Spectrosc Radiat Transfer, 2016,183, 128-153;b) Kawabata, K. 15-digit accuracy calculations of Ambartsumian-         Chandrasekhar`s H-functions for four-term phase functions with double- exponential formula. Astrophysics and Space Science, 2017, vol.363, Issue 1. In line 67 of the manuscript there is a sentence “Essentially, the integral equation in (1) has a set of solutions”. But Eq. (1) is an integro-differential equation, not an integral equation. In addition, only one of the boundary conditions is an integral relation, but not an integral equation. The integration domain should be pointed out in Eq.(6) In Eq. (14) the symbol of the ordinary derivative should be replaced by the symbol of the partial derivative. In the manuscript the selection problem of the values of non-negative integers M, K, N is discussed very briefly and not very convincingly For a potential reader of the work, it would be useful to briefly explain equality (46) or make a reference to the work where it was first used.

    9. The lack of comparison of the obtained approximate solutions of BVPs          for RTE with exact or asymptotic solutions of the same problems (at   least for special cases)is a significant drawback of the work. The authors of the manuscript in their previous publications compared their approximate solutions only with other approximate solutions.  Therefore, the true accuracy of these decisions is very difficult or impossible to evaluate  in this way. Moreover, the authors did not use rigorous a priori or a posteriori estimates obtained on the basis of functional analysis. In line 310 of the manuscript it is asserted that MDOM provides average convergence. This general statement requires substantiation using some sufficient conditions when such convergence takes place.10. In line 201, the word “Ambardzumyan” should be replaced by the name “Ambartsumian”.11. In line 200 a phrase  «which allows us to derive the invariant imbedding method V.A. Ambardzumyan [16]» is an incorrect phrase. After all, the authors only established a formal connection between their results and the Ambartsumian method, and did not propose this method themselves.12. In line 228 it is asserted that  «the invariance principle can be applied to a layer of arbitrary vertical inhomogeneity». But this fact has been used in RTT by other famous researchers for several decades. However, the authors do not provide any references to their articles and monographs. In line 228 of the text of the work, the authors use the term “The invariance principle ...”. However, a rather large number of such principles have been formulated in RTT and theoretical physics. Therefore, the text in line 228 should be clarified and specified.

In Eq.(37) there is a misprint.

  14. In  lines 369-371 of the manuscript text it is asserted that  in the case of a three-dimensional medium (3D) BVPs for RTE must be solved numerically. This statement is only partially true. A large array of exact and asymptotic solutions of BVPs for RTE for three-dimensional media cases has already been obtained in the scientific literature.  However this part of the achievements was almost completely ignored by the authors of the manuscript  although it was announced in Abstract that they would review and analyze the current state of the radiative transfer theory (RTT).

 15. The first sentence in Conclusions (see lines 434, 435) does not reflect the essence of the material presented in the manuscript. In fact, the authors presented in the manuscript, basically, only their results (of the 33 works in the list of references, 14 publications belong to the authors of the manuscript), which were partially published by them earlier. Therefore, the authors failed to fully demonstrate the harmony and beauty that the RTT has achieved in more than 100 years.

 16. In lines 107, 108, there is a reference to [7]. But similar, generally speaking, results concerning the formulation of boundary value problems for RTE and the role of the Green function were also obtained in the monograph Case K.M., Zweifel P.F. (1967).